# `Q3R`: Quadratic Reweighted Rank Regularizer for Effective Low-Rank Training

**Ipsita Ghosh**\*
Department of Computer Science
University of Central Florida
ipsita.ghosh@ucf.edu

**Ethan Nguyen**\*
Department of Computer Science
University of North Carolina at Charlotte
ethan.nguyen@e-10.net

**Christian Kümmerle**
School of Data, Mathematical and Statistical Sciences
Department of Computer Science
University of Central Florida
kuemmerle@ucf.edu

## Abstract

Parameter-efficient training, based on low-rank optimization, has become a highly successful tool for fine-tuning large deep-learning models. However, these methods fail at low-rank pre-training tasks where maintaining the low-rank structure and the objective remains a challenging task. We propose the *Quadratic Reweighted Rank Regularizer* dubbed `Q3R`, which leads to a novel low-rank inducing training strategy inspired by the iteratively reweighted least squares (IRLS) framework. `Q3R` is based on a quadratic regularizer term which majorizes a smoothed log determinant serving as rank surrogate objective. Unlike other low-rank training techniques, `Q3R` is able to train weight matrices with prescribed, low target ranks of models that achieve comparable predictive performance as dense models, with small computational overhead, while remaining fully compatible with existing architectures. In experiments, we are able to truncate 60% of the parameters of a ViT-Tiny parameters with marginal loss in CIFAR-10 performance and up to 80% with only 4% accuracy drop. The efficacy of `Q3R` is confirmed on Transformers across both image and language tasks, including for low-rank fine-tuning.

The code is available at https://github.com/ThatE10/q3r.git.

## 1 Introduction

Modern deep learning architectures continue to grow in size and complexity (RWC+19), creating a growing demand for efficient training methodologies. Low-rank regularization has emerged as a powerful paradigm for addressing these challenges by explicitly constraining the parameter search space through matrix factorization. This approach builds on the empirical observation that neural networks exhibit inherent low-dimensional structure in their weight matrices during training (GK+22).

Practical implementations face three key challenges: (1) performance degradation compared to full-rank baselines, (2) optimal rank selection across layers, and (3) maintaining training stability. Prior work addresses these through spectral initialization (GK+22), orthogonality regularization (YYT+20).

---

\*Equal contribution.

39th Conference on Neural Information Processing Systems (NeurIPS 2025).

Recent advances in parameter-efficient fine-tuning (PEFT) have expanded the low-rank training paradigm through methods like Low-Rank Induced Training (LoRITa) (AZW). These approaches maintain the original model architecture during inference while inducing low-rank structure through strategic layer overparameterization during training. LoRITa specifically decomposes weight matrices $\mathbf{W}_i$ into products $\prod_{k=1}^{N} \mathbf{W}_i^k$ during optimization, enabling implicit rank reduction through singular value truncation post-training (AZW). This methodology demonstrates that explicit rank constraints can be replaced by training dynamics that naturally favor low-rank solutions.

Despite their promise, existing low-rank training approaches present several notable limitations. Traditional low-rank methods often suffer from performance degradation relative to full-rank baselines (GK$^+$22; YYT$^+$20). Methods such as LoRA and LoRITa, while effective at reducing trainable parameters, can struggle to capture the full structure making it difficult to generalize to complex tasks (AZW). Furthermore, PEFT techniques introduce additional hyperparameters (such as rank and scaling factors) whose optimal values may not generalize across architectures, datasets, or downstream tasks, often requiring extensive re-tuning and experimentation. In multilingual or low-resource settings, PEFT methods like LoRA have been observed to yield inconsistent results, sometimes improving language-specific generation at the expense of reasoning or generalization abilities (KJP$^+$25). Combining multiple PEFT modules for multi-task or continual learning can also lead to increased memory usage and system complexity, offsetting some of the intended efficiency gains. Overall, it can be observed that the advances in LoRA-type parameter-efficient training methods have not yet been able to be translated to enable robust low-rank pre-training.

## 2 Contribution

We propose **Quadratic Reweighted Rank Regularization** (Q3R), which solves the pre-existing problems by introducing an optimizer-compatible regularization framework based on smoothed log-determinant rank surrogates outlined in Section 4.1, which is specifically designed for low-rank pre-training. Our approach is theoretically grounded in saddle-escaping second-order optimization methods, and it comes with little computational overhead compared to unregularized training despite its efficacy for promoting low-rank neural network weight matrices. Additionally, we propose the Adam variant AdamQ3R in Section 4.2, which is tailored to optimizing Q3R-regularized loss functions and which improves the performance of training Q3R-regularized models.

Numerical experiments show that Q3R is able to reduce the number of parameters in ViT models by $60\%$ during pre-training on CIFAR-10, with only around $1.3\%$ accuracy drop. We validate the performance of Q3R for low-rank fine-tuning with experiments fine-tuning RoBERTa and Llama3 on GLUE tasks, for which Q3R achieves comparable performance compared to dense fine-tuning and state-of-the art low-rank PEFT methods. Compared to state-of-the-art low-rank training methods such as LoRA (HSW$^+$22b), LoRITa (AZW), Q3R consistently produces models with better generalization at high truncation levels, without requiring overparameterization or full-rank warmup phases.

In Section 4, we elaborate the methodology of the proposed work, which is further discussed with a detailed derivation in the Supplementary material in Appendix A. In Section 5, we empirically show the performance of Q3R, in comparison to other state-of-the-art methods. We continue with more experimental evaluations in the Supplementary material in Appendix D. Appendix D also includes discussions of the computational aspects of our methodology. In Appendix E, we demonstrate the robustness of Q3Rto different hyperparameter variations. We briefly discuss the limitations of our work in Section 6.

## 3 Related Work

**Parameter-Efficient Fine-Tuning (PEFT)** Parameter-efficient fine-tuning is the concept of modifying only parts of a fully parametrized pre-trained model to excel at specific task of interest. PEFT methods such as adapters (HGJ$^+$19) and LoRA (HSW$^+$22b) introduce small, trainable modules into a frozen pretrained model, drastically reducing the number of parameters to be updated. These techniques often match full fine-tuning performance with only a tiny fraction of trainable parameters. However, the low-rank constraints that make LoRA-style methods efficient for downstream tasks also limit capacity if applied during pre-training. Training from scratch with only low-rank adapters or factorizations (instead of full-rank weight updates) tends to underperform, as it restricts

optimization to a low-dimensional subspace (ZZC$^+$24). LoRA assumes a well-formed pretrained weight $W$ plus a low-rank perturbation of rank-$r$ adapter matrices $A \in \mathbb{R}^{d \times r}$ and $B \in \mathbb{R}^{r \times d}$ such that $\Delta W = AB$; without a strong initial $W$, such updates struggle to capture the full complexity needed for learning from scratch. *KronA* replaces LoRA's product with a Kronecker factorization for better rank-parameter trade-offs (ETK$^+$25). *DoRA* decouples update magnitude and direction via a learnable scaling factor, improving upon LoRA's expressivity (LWY$^+$24). *Compacter* uses shared, low-rank, Kronecker-parameterized adapters across layers, matching standard adapters with only 0.05 % extra parameters (MHR21).

**Low-Rank Training in Neural Networks.**   Neural networks often exhibit implicit low-rank structure during training, as optimization dynamics like SGD with weight decay tend to bias models toward low-rank solutions (GSGP25; HMZ$^+$23). This observation has motivated a range of explicit low-rank training methods that constrain parameter matrices directly. A common approach factorizes weights and trains the factorized weights instead, reducing compute and memory costs with minor accuracy loss (KTMF21). Techniques like LoRA (WMPG24b) and its extensions (LSMR23) inject low-rank updates into pretrained Transformer weights, enabling parameter-efficient adaptation. However, pre-training directly under low-rank constraints remains more challenging. (WMPG24a) shares a similar motivation to ours—studying the limitations of LoRA-style low-rank pre-training and proposing an alternative regularization-driven approach to induce low-rank structure during training. Although we approach the problem through a different optimization framework, their analysis and framing of the limitations of adapter-based methods are highly relevant and can guide refinement of both the positioning and justification of our method. Regularization-based approaches use nuclear norm or log-determinant surrogates to promote low-rank solutions (SZCT23), while others apply orthogonality constraints and adaptive rank pruning (YYT$^+$20; YCS$^+$20). In Transformers, low-rank parameterizations have achieved 2–5× compression with minimal performance drop (AZW), and Cuttlefish (WAUc$^+$23) automates rank selection by monitoring stable ranks during a warmup phase. Still, many methods rely on post-hoc truncation or overparameterization, which do not minimize rank during training. Our work addresses this gap by directly optimizing for low-rank solutions via reweighted least squares, promoting compact representations throughout pre-training. However, many of these methods rely on overparameterization or post-hoc truncation and do not directly minimize rank during training. In contrast, our approach promotes low-rank structure directly via optimization, using a principled regularization technique rooted in reweighted least squares.

**Spectral Low-Rank Regularization.**   A related line of work studies algorithms that impose low-rankness of neural network matrices based on the nuclear norm, Schatten-$p$ quasi-norm or a direct rank regularization. In particular, (AS17) proposed a proximal stochastic gradient descent applied to the nuclear norm. Methods that apply spectral truncation (e.g., via truncated SVD) during training or post-training (YTW$^+$20; XLZ$^+$20) can also be understood within this framework. A downside of such approaches is the computational overhead: they require at least a truncated SVD at *every* iteration, which quickly becomes computationally prohibitive for larger networks. Moreover, arguably, such aggressive, discontinuous rank regularization interferes with the continuous gradient-based training process of the network.

In contrast, our proposed `Q3R` regularizer imposes low-rankness more *gradually* by reweighting at periodic intervals, at which a smoothing parameter is updated as well, which we find to be sufficient for convergence while significantly reducing the computational overhead. This "soft" imposition of low-rank structure aligns with insights from IRLS-based methods (see below). A spectral, Schatten-$p$ regularization is also the core of the motivation of LoRITa (AZW); however, in this case, the spectral regularization can be seen as a justification of an (unweighted) squared Frobenius norm regularization on factor matrices, whereas `Q3R` does not work with factor matrices and considers a *reweighted* quadratic term.

**Rank Regularization and IRLS.**   In a line of work that significantly precedes the interest in low-rank techniques for deep learning, the problem of identifying or learning a low-rank matrix from noisy, under-determined linear measurements has been studied for decades in control theory (FHB03; DCM22), recommender systems (Kor09; KBV09) and compressed sensing (RFP10; DR16). Even in this setting, which is fundamentally linear unlike the training of deep neural networks, the minimization of a rank objective subject to the constraints is NP-hard (Nat95; RFP10) , motivating surrogate formulations or relaxations. Convex relaxations using the nuclear norm (CR09; RFP10;

DR16; CW18) have been popular for a long time due to their strong recovery guarantees under suitable assumptions and their ability to be tackled using the machinery of convex optimization (BV04), but fail to result in a convex formulation in the deep learning setting. Even disregarding computational limitations of convex regularizations (CC18), *non-convex* rank surrogates such as the log-determinant penalty (FHB03) lead to algorithms which are more data-efficient as a evidenced for a variety of low-rank matrix recovery problems (FHB04; CESV13; KS18; KM23).

If combined with a suitable smoothing strategy, the non-smooth optimization framework of Iteratively Reweighted Least Squares (IRLS), originally pioneered by Weiszfeld (Wei37; WP09; BS15), has emerged as a leading algorithmic framework to optimize non-convex rank surrogates (FRW11; MF10; KMV21; GTK24) as it provides good trade-offs between scalability, data-efficiency, saddle-point evasion (present due to inherent non-convexity) and fast convergence. On a high level, IRLS solves a sequence of weighted Frobenius-norm problems that progressively suppress smaller singular values. The proposed rank regularization term `Q3R` (detailed in Section 4.1) builds on recent improvements on low-rank IRLS weight operator formulations (KMV21; GTK24) (or reweighting strategies), which, unlike older formulations (FRW11; MF10), allow for fast saddle-point evasion and locally quadratic convergence rates. To the best of our knowledge, IRLS-type low-rank regularization, which is at the core of `Q3R`, has not been explored in the literature in the context of deep learning so far. While providing an interesting perspective on older IRLS formulations (FRW11; MF10) from an *average gradient outer product* perspective, the recent work (RBD25) does not provide insights towards the derivation of quadratically convergent IRLS methods (KMV21; GTK24), nor does it extend the framework towards low-rank training of deep networks.

In the language of the low-rank recovery literature, LoRA-type (HSW$^+$22a) approaches are known under the name of (*Burer-Monteiro* (BM03)) matrix factorization methods (SL16; ZL15; MWCC20; CLC19; ZCZ22; XSCM23; SZ25). While popular in applications (KBV09; RKZK22) due to their scalability, it is known that they can be outperformed by IRLS or Riemannian optimization approaches in more challenging setups involving, e.g., limited data (ZN22; LHLZ24), which is one motivation of our work.

## 4    Methodology

In this section, we provide a detailed derivation and definition of the *Quadratic Reweighted Rank Regularizer* `Q3R` in Section 4.1, before we embed it into a training scheme to train low-rank weights of deep learning models in Section 4.2.

### 4.1    Low-Rank Regularization via `Q3R`

Given a neural network with $K$ weight matrices $\Theta = \{\mathbf{W}_i : \mathbf{W}_i \text{ is weight matrix}, i = 1, \dots, K\}$, a functionally ideal regularization term to add to the loss function of the network for the promotion of a low-rank weight is simply the *rank* of $\mathbf{W}$. However, $\mathrm{rank}(\mathbf{W})$ is non-convex and not continuous, and thus hard to incorporate into a gradient-based training methodology.

In the following, we consider the *non-convex*, but continuously differentiable rank surrogate $F_\epsilon(\cdot)$ called *$\epsilon$-smoothed log-determinant*, defined as

$$F_\epsilon(\mathbf{W}) := \sum_{i=1}^{d} f_\epsilon(\sigma_i(\mathbf{W})), \text{ where } f_\epsilon(\sigma) = \begin{cases} \epsilon^2 \left(\log(\sigma) - \log(\epsilon)\right) + \frac{1}{2}\epsilon^2, & \text{if } \sigma \geqslant \epsilon, \\ \frac{1}{2}\sigma^2, & \text{if } \sigma < \epsilon, \end{cases} \quad (1)$$

where $\sigma_i(\mathbf{W})$ is the $i$-th singular value of $\mathbf{W}$, and which is parametrized by a smoothing parameter $\epsilon > 0$. The definition of (1) follows (KM23) and is related to the log-determinant heuristics $\log \det(\mathbf{W} + \epsilon I) = \sum_{i=1}^{d} \log(\sigma_i(\mathbf{W}) + \epsilon)$ defined for positive semi-definite matrices $\mathbf{W} \in \mathbb{R}^{d \times d}$ of (FHB03; CESV13; RBD25). Compared to other log-determinant type functions, $F_\epsilon(\cdot)$ from (1) has a few advantages: It is lower bounded by 0 for any $\epsilon$ and has a 1-Lipschitz gradient, making the objective compatible with gradient-based optimizers without extensive step-size adaptation. Furthermore, its smoothing parameter $\epsilon$ *regulates* the non-convexity of its optimization landscape and recovers the well-known squared Frobenius norm as $F_\epsilon(\mathbf{W}) = \frac{1}{2}\|\mathbf{W}\|_F^2$ in the case of $\sigma_1(\mathbf{W}) < \epsilon$.

The rank regularizer we study in this paper, however, is not simply $F_\epsilon(\cdot)$: If we were to work directly with the $\epsilon$-smoothed log-determinant, its gradients $\nabla F_\epsilon(\mathbf{W})$ would require a full spectral decomposition of $\mathbf{W}$ at each training iteration (see supplementary material).

Instead, we consider, given an expansion center point $\mathbf{W}'$ (which may correspond to the current weight matrix during a neural network training dynamics), the *quadratic model* $Q_\epsilon(\cdot|\mathbf{W}')$ defined as

$$Q_\epsilon(\mathbf{W} \mid \mathbf{W}') = F_\epsilon(\mathbf{W}') + \tfrac{1}{2}\langle \mathbf{W}, \mathcal{R}_{\mathbf{W}',\epsilon}(\mathbf{W})\rangle - \tfrac{1}{2}\langle \mathbf{W}', \mathcal{R}_{\mathbf{W}',\epsilon}(\mathbf{W}')\rangle, \tag{2}$$

where $\mathcal{R}_{\mathbf{W}',\epsilon}(\cdot) : \mathbb{R}^{d_1 \times d_2} \to \mathbb{R}^{d_1 \times d_2}$ is a positive definite, so-called *reweighting operator* (KMV21; GTK24), defined in Definition 4.1 below.

**Definition 4.1** (Reweighting Operator (KM23))**.** *Let $\epsilon > 0$ and $\mathbf{W}' \in \mathbb{R}^{d_1 \times d_2}$ be a matrix with singular value decomposition $\mathbf{W}' = \mathbf{U}_{\mathbf{W}'} \operatorname{diag}(\sigma_i(\mathbf{W}'))\mathbf{V}_{\mathbf{W}'}^\top$, where $\mathbf{U} \in \mathbb{R}^{d_1 \times r(\mathbf{W}',\epsilon)}$ and $\mathbf{V} \in \mathbb{R}^{d_2 \times r(\mathbf{W}',\epsilon)}$ are matrices of the leading $r(\mathbf{W}',\epsilon)$ left and right singular vectors satisfying $\mathbf{U}_{\mathbf{W}'} = [\mathbf{U} \quad \mathbf{U}_\perp] \in \mathbb{R}^{d_1 \times d_1}$ and $\mathbf{V}_{\mathbf{W}'} = [\mathbf{V} \quad \mathbf{V}_\perp] \in \mathbb{R}^{d_2 \times d_2}$, and*

$$r(\mathbf{W}',\epsilon) := |\{i \in \{1,\ldots,\min(d_1,d_2)\} : \sigma_i(\mathbf{W}') > \epsilon\}| \tag{3}$$

*is the number of singular values of $\mathbf{W}'$ larger than $\epsilon$. Then we define the* reweighting operator *$\mathcal{R}_{\mathbf{W}',\epsilon} : \mathbb{R}^{d_1 \times d_2} \to \mathbb{R}^{d_1 \times d_2}$ associated to the matrix $\mathbf{W}'$ and smoothing parameter $\epsilon$ as*

$$\mathcal{R}_{\mathbf{W}',\epsilon}(\mathbf{W}) = \mathbf{U}_{\mathbf{W}'}\Sigma_{\epsilon,d_1}^{-1}\mathbf{U}_{\mathbf{W}'}^\top \mathbf{W}\mathbf{V}_{\mathbf{W}'}\Sigma_{\epsilon,d_2}^{-1}\mathbf{V}_{\mathbf{W}'}^\top,$$

*where $\Sigma_{\epsilon,d} = \operatorname{diag}(\max(\sigma_i(\mathbf{W}')/\epsilon, 1))_{i=1}^d \in \mathbb{R}^{d \times d}$ for $d \in \{d_1, d_2\}$.*

The reweighting operator satisfies the following simple properties (shown in the supplementary material), which makes working with it computationally feasible.

**Lemma 4.1.** *For $\epsilon > 0$ and $\mathbf{W}'$, let $\mathbf{U} \in \mathbb{R}^{d_1 \times r(\mathbf{W}',\epsilon)}$, $\mathbf{V} \in \mathbb{R}^{d_1 \times r(\mathbf{W}',\epsilon)}$ and $\mathcal{R}_{\mathbf{W}',\epsilon} : \mathbb{R}^{d_1 \times d_2} \to \mathbb{R}^{d_1 \times d_2}$ be as in Definition 4.1. Then the following statements are true:*

*1. $\mathcal{R}_{\mathbf{W}',\epsilon}(\cdot)$ is a positive definite operator with respect to the Frobenius inner product $\langle \mathbf{A}, \mathbf{B}\rangle = \operatorname{tr}(\mathbf{A}^\top\mathbf{B})$, i.e., $\langle \mathbf{W}, \mathcal{R}_{\mathbf{W}',\epsilon}(\mathbf{W})\rangle > 0$ for all non-zero $\mathbf{W} \in \mathbb{R}^{d_1 \times d_2}$.*

*2. The image $\mathcal{R}_{\mathbf{W}',\epsilon}(\mathbf{W})$ of any $\mathbf{W} \in \mathbb{R}^{d_1 \times d_2}$ w.r.t. the reweighting operator can be computed as*

$$\begin{aligned}
\mathcal{R}_{\mathbf{W}',\epsilon}(\mathbf{W}) = {} & \epsilon^2\mathbf{U}\Sigma^{-1}\mathbf{U}^\top\mathbf{W}\mathbf{V}\Sigma^{-1}\mathbf{V}^\top + \epsilon\mathbf{U}\Sigma^{-1}\mathbf{U}^\top\mathbf{W}(\mathbf{I}-\mathbf{V}\mathbf{V}^\top) \\
& + \epsilon(\mathbf{I}-\mathbf{U}\mathbf{U}^\top)\mathbf{W}\mathbf{V}\Sigma^{-1}\mathbf{V}^\top + (\mathbf{I}-\mathbf{U}\mathbf{U}^\top)\mathbf{W}(\mathbf{I}-\mathbf{V}\mathbf{V}^\top),
\end{aligned} \tag{4}$$

*where $\Sigma = \operatorname{diag}(\sigma_i(\mathbf{W}'))_{i=1}^{r(\mathbf{W}',\epsilon)} \in \mathbb{R}^{r(\mathbf{W}',\epsilon) \times r(\mathbf{W}',\epsilon)}$ is the diagonal matrix containing the largest $r(\mathbf{W}',\epsilon)$ singular values of $\mathbf{W}'$.*

*3. The quadratic model of (2) satisfies, for all $\mathbf{W}, \mathbf{W}' \in \mathbb{R}^{d_1 \times d_2}$, that*

$$Q_\epsilon(\mathbf{W} \mid \mathbf{W}') = F_\epsilon(\mathbf{W}') + \langle \nabla F_\epsilon(\mathbf{W}'), \mathbf{W} - \mathbf{W}'\rangle + \tfrac{1}{2}\langle \mathbf{W} - \mathbf{W}', \mathcal{R}_{\mathbf{W}',\epsilon}(\mathbf{W} - \mathbf{W}')\rangle. \tag{5}$$

We note that the quadratic model $Q_\epsilon(\cdot \mid \mathbf{W}')$ (2) defined by $\mathcal{R}_{\mathbf{W}',\epsilon}$ is a *majorizing* quadratic model that satisfies $Q_\epsilon(\mathbf{W}|\mathbf{W}') \geqslant F_\epsilon(\mathbf{W})$ for all $\mathbf{W} \in \mathbb{R}^{d_1 \times d_2}$.[2] It is *different* from a second-order Taylor expansion of the $F_\epsilon(\cdot)$ about $\mathbf{W}'$, which would only be an *approximation, but no majorization* due to the non-convex nature of $F_\epsilon(\cdot)$. The quadratic model can still be related to a second-order Taylor expansion of $F_\epsilon$ via (5) as each generalized Hessian (HUSN84) $\partial^2 F_\epsilon(\mathbf{W}')$ of $F_\epsilon$ satisfies $\partial^2 F_\epsilon(\mathbf{W}') \preceq \mathcal{R}_{\mathbf{W}',\epsilon}$ in the Loewner order.

We observe that in the quadratic model $Q_\epsilon(\mathbf{W} \mid \mathbf{W}')$ of (2), the only term that depends on $\mathbf{W}$ is the second summand. Thus, to obtain a simple, *differentiable* regularizer term that can be incorporated into a deep learning framework, we define the *Quadratic Reweighted Rank Regularizer* Q3R of a neural network weight matrix $\mathbf{W} \in \mathbb{R}^{d_1 \times d_2}$, given $\mathbf{W}' \in \mathbb{R}^{d_1 \times d_2}$ and $\epsilon > 0$, as $\text{Q3R}_{\mathbf{W}',\epsilon} : \mathbb{R}^{d_1 \times d_2} \to \mathbb{R}$ with

$$\text{Q3R}_{\mathbf{W}',\epsilon}(\mathbf{W}) = \frac{1}{2}\langle \mathbf{W}, \mathcal{R}_{\mathbf{W}',\epsilon}(\mathbf{W})\rangle. \tag{6}$$

As we see in the next section, it is simple and tractable to compute its gradient $\nabla_{\mathbf{W}} \text{Q3R}_{\mathbf{W}',\epsilon}(\mathbf{W}) \in \mathbb{R}^{d_1 \times d_2}$, which can be used by any gradient-based optimizer.

---

[2]This majorization property is implicitly postulated in (KMV21; KM23), but without proof. While proving this property is beyond the scope of this paper, we believe that the statement is true.

**Algorithm 1** Update Reweighting Operator $\mathcal{R}_{\mathbf{W}',\epsilon_{\text{old}}}(\cdot) \mapsto \mathcal{R}_{\mathbf{W},\epsilon_{\text{new}}}(\cdot)$

1: **Input:** NN weight matrix $\mathbf{W} \in \mathbb{R}^{d_1 \times d_2}$; target rank $r_{\text{target}}$; prev. smoothing parameter $\epsilon_{\text{old}}$.
2: **Output:** Updated $\epsilon_{\text{new}}$, reweighting operator $\mathcal{R}_{\mathbf{W},\epsilon_{\text{new}}}$ (via $\Sigma$, $\mathbf{U}$, $\mathbf{V}$), envelope rank $r_{\text{env}}$
3: Compute $[\mathbf{U},\Sigma,\mathbf{V}] = \text{SVD}^{\epsilon_{\text{old}}}(\mathbf{W})$ of $\mathbf{W}$, where $\text{SVD}^{\epsilon_{\text{old}}}(\cdot)$ computes a partial singular value decomposition of its input up to order $r(\cdot, \epsilon_{\text{old}})$ (see (3)) as well as $\sigma_{r_{\text{target}}+1}(\mathbf{W})$.
4: $\epsilon_{\text{new}} = \min\left(\epsilon_{\text{old}}, \sigma_{r_{\text{target}}+1}(\mathbf{W})\right)$.      ▷ UPDATE SMOOTHING (7)
5: $r_{\text{env}} = r(\mathbf{W}, \epsilon_{\text{new}})$      ▷ UPDATE RANK ENVELOPE
6: Set $\mathbf{U} = \mathbf{U}_{:,1:r_{\text{env}}}, \mathbf{V} = \mathbf{V}_{:,1:r_{\text{env}}}, \Sigma = \Sigma_{1:r_{\text{env}},1:r_{\text{env}}}$      ▷ RESTRICT PART. SVD MATRICES
7: **return** Reweighting operator $\mathcal{R}_{\mathbf{W},\epsilon_{\text{new}}}$ implicitly defined by $\mathbf{U} \in \mathbb{R}^{d_1 \times r_{\text{env}}}, \mathbf{V} \in \mathbb{R}^{d_2 \times r_{\text{env}}}, \Sigma$ & $\epsilon_{\text{new}}$.

Furthermore, we periodically (but not at each training iteration) *update* the reweighting operator of Q3R (and thus, the underlying quadratic model $Q_\epsilon(\mathbf{W} \mid \mathbf{W}')$) by setting $\mathbf{W}' \leftarrow \mathbf{W}$ and re-compute $\mathcal{R}_{\mathbf{W}',\epsilon}$, for which a truncated SVD of $\mathbf{W}$ is sufficient due to (4). Additionally, whenever updating $\mathcal{R}_{\mathbf{W}',\epsilon}$, we apply the *non-increasing* update

$$\epsilon \leftarrow \min(\epsilon, \sigma_{r_{\text{target}}+1}(\mathbf{W})) \tag{7}$$

to the smoothing parameter $\epsilon$, which uses a *target rank* parameter $r_{\text{target}}$ as an input. The rationale of this smoothing parameter update is two-fold: first, this choice gives *partial control* on the expected rank of the weight matrix after training, as the value of $\mathcal{R}_{\mathbf{W}',\epsilon}(\mathbf{W})$ tends to 0 if $\epsilon$ follows the dynamics of (7) in the case of $\epsilon \to 0$ for matrices $\mathbf{W}$ whose row and column spaces are both orthogonal to the columns of $\mathbf{U}$ and $\mathbf{V}$, respectively. Second, this choice increases the *non-convexity* the $\epsilon$-smoothed log-determinant $F_\epsilon$ underlying Q3R *gradually* (YATC20), facilitating a fast convergence to true low-rank solutions without becoming trapped in high-rank local minima (KMV21; KM23). We summarize computational steps of a reweighting operator update in Algorithm 1.

### 4.2 Neural Network Training via AdamQ3R

Let now $y : \mathbb{R}^{d_1^1 \times d_2^1} \times \mathbb{R}^{d_1^K \times d_2^K} \times \mathbb{R}^{d_{\text{in}}} \to \mathbb{R}^{d_{\text{out}}}$ be the input-output mapping of a deep neural network that depends on weight parameter matrices $\Theta = \{\mathbf{W}_k \in \mathbb{R}^{d_1^k \times d_2^k} : \mathbf{W}_k \text{ is weight matrix}, k = 1, \ldots, K\}$. For a Transformer-based architecture such as Vision Transformer (DBKea21), the weight matrices include square layer- and head-wise query, key and value weight matrices $\mathbf{W}_q, \mathbf{W}_k, \mathbf{W}_v \in \mathbb{R}^{d \times d}$ as well as rectangular projection and MLP layer weight matrices. Given a pairwise loss $\ell(\cdot, \cdot)$ such as cross entropy and a training dataset $\{x_i, y_i\}_{i=1}^n$, we can define the (unregularized) network loss as $\mathcal{L}(\Theta) = \frac{1}{n}\sum_{i=1}^n \ell(y(\Theta, x_i), y_i)$.

In order to gradually impose low-rank weights during training, we propose to optimize instead the Q3R-regularized total loss $\mathcal{L}_{\text{Q3R}}(\Theta) := \mathcal{L}(\Theta) + \lambda \sum_{k=1:\text{Q3R is active for } \mathbf{W}_k}^K \text{Q3R}_{\mathbf{W}_k',\epsilon_k}(\mathbf{W}_k)$, where $\lambda > 0$ is a regularization parameter and the $\text{Q3R}_{\mathbf{W}_k',\epsilon_k}(\cdot)$ are as in (6); the $\{\mathbf{W}_k'\}$ are initially set to the initialization weights, and $\epsilon_k = \infty$ for each $k = 1, \ldots, K$. We observe that due to the definition of Q3R, the gradient with respect to the regularizer terms can be computed as $\nabla_{\mathbf{W}_k} \text{Q3R}_{\mathbf{W}_k',\epsilon_k}(\mathbf{W}_k) = \mathcal{R}_{\mathbf{W}_k',\epsilon_k}(\mathbf{W}_k)$ for each $k$, i.e., by computing the image of $\mathbf{W}_k$ with respect to the reweighting operator $\mathcal{R}_{\mathbf{W}_k',\epsilon_k}(\cdot)$ of Definition 4.1.

The Q3R-regularized loss $\mathcal{L}_{\text{Q3R}}$ can now be used in conjunction with any optimizer suitable for the neural network architecture such as minibatch stochastic gradient or Adam (A$^+$14). To ensure that the quadratic models underlying Q3R match the $\epsilon_k$-smoothed log-determinant rank surrogates

---

**Algorithm 2** Low-Rank Training via AdamQ3R

**Input:** Minibatch size $B$, reweighting period $T$, Q3R parameter $\lambda$, learning rate $\alpha = 0.001$, $\beta_1 = 0.9$, $\beta_2 = 0.999$, $\delta = 10^{-8}$, $\eta = 3$, target rank $r_{\text{target}}$.
1: Initialize parameter $\mathbf{W}_0$, $\epsilon_0$ and reweighting operator $\mathcal{R}_{\mathbf{W}_0,\epsilon_0}$
2: **for** $t = 0, 1, \ldots$ **do**
3:   **if** $t \mod T = 0$ **then**
4:     Update reweighting operator $\mathcal{R}_{\lfloor \frac{t}{T} \rfloor}(\cdot) := \mathcal{R}_{\mathbf{W}_t,\epsilon_t}(\cdot)$ and $\epsilon_t$      ▷ USE ALGORITHM 1
5:   **end if**
6:   Sample minibatch $S = \{(x_i, y_i)\}_{i=1}^B$
7:   $\mathbf{g}_{t+1} \leftarrow \nabla_{\mathbf{W}} \mathcal{L}_S(\mathbf{W}_t)$      ▷ COMPUTE BATCH GRADIENT OF $\mathcal{L}$
8:   $\mathbf{m}_{t+1} \leftarrow \beta_1 \mathbf{m}_t + (1 - \beta_1)\mathbf{g}_{t+1}$
9:   $\mathbf{v}_{t+1} \leftarrow \beta_2 \mathbf{v}_t + (1 - \beta_2)\mathbf{g}_{t+1}^2$
10:   $\hat{\mathbf{m}}_{t+1} \leftarrow \mathbf{m}_{t+1}/(1 - \beta_1^{t+1})$
11:   $\hat{\mathbf{v}}_{t+1} \leftarrow \mathbf{v}_{t+1}/(1 - \beta_2^{t+1})$
12:   $\mathbf{R}_t \leftarrow \mathcal{R}_{\lfloor \frac{t}{T} \rfloor}(\mathbf{W}_t)$ ▷ COMPUTE Q3R GRADIENT (4)
13:   $\mathbf{W}_{t+1} \leftarrow \mathbf{W}_t - \eta\left(\frac{\alpha \hat{\mathbf{m}}_{t+1}}{\sqrt{\hat{\mathbf{v}}_{t+1}} + \delta} + \lambda \mathbf{R}_t\right)$
14: **end for**
15: **return** $\mathbf{W}_t$

$F_{\epsilon_k}$ (1) closely, we update the reweighting operators $\mathcal{R}_{\mathbf{W}'_k, \epsilon_k}(\cdot)$ via Algorithm 1 on a fixed iteration schedule of every $T$ training iterations–we call this parameter $T$ the *reweighting period*.

However, instead of using a generic adaptive gradient optimizer such as Adam on $\mathcal{L}_{\texttt{Q3R}}$, we observe that the Q3R terms already possess accurate second-order information of underlying regularization surrogate, which means that including the Q3R terms into the adaptive part of an Adam-like optimizer is likely to be suboptimal. For this reason, we propose to use a dedicated adaptive optimizer to optimize $\mathcal{L}_{\texttt{Q3R}}$, dubbed *AdamQ3R*, which is detailed in Algorithm 2. AdamQ3R extends the observation of AdamW (LH19) that a decoupling of regularization term (in that case, squared Frobenius norm regularization) and network loss improves generalization performance to Q3R regularization, avoiding a distortion of the loss landscape. A validation of the benefits of using AdamQ3R vs. standard Adam applied to $\mathcal{L}_{\texttt{Q3R}}$ can be found in the supplementary material.

**Computational Aspects.** Following the low-rank training framework of Q3R, for example, via AdamQ3R, introduces a limited computational overhead compared to unregularized deep learning. Every $T$ training iterations, a truncated singular value decomposition of order $r_{\text{env}}$ (see Algorithm 1) of each weight matrix $\mathbf{W}_i \in \mathbb{R}^{d_1 \times d_2}$ to which Q3R is applied is required, which has a time complexity of $O(d_1 d_2 r_{\text{env}} + (d_1 + d_2) r_{\text{env}}^2)$ (HMT11). Similarly, calculating a Q3R gradient $\mathbf{R}_t$ in Algorithm 2 imposes a total cost of $O(d_1 d_2 r_{\text{env}} + (d_1 + d_2) r_{\text{env}}^2 + r_{\text{env}}^3)$. Since the smoothing parameter update rule (7) is designed to relate $r_{\text{env}}$ with the target rank $r_{\text{target}}$ such that $r_{\text{env}} \approx r_{\text{target}}$, the additional time complexity is somewhat proportional to the target rank. To obtain significant parameter reductions in the trained network weight matrices, it is chosen such that $r_{\text{target}} \ll \min(d_1, d_2)$, which limits the computational overhead of Q3R in the most interesting use cases. Additional memory requirements amount to $r_{\text{env}}(d_1 + d_2 + 1)$ per weight matrix as the reweighting operator information needs to be stored via $\mathbf{U}, \mathbf{V}$ and $\Sigma$.

# 5 Experiments

We explore the ability of Q3R to obtain favorable trade-offs between model performance and parameter-efficiency across diverse architectures and data distributions experimentally. To this end, we compare different low-rank training methodologies across a range of architecture-dataset pairs: we pre-train ViT-Tiny (DBKea21; SKZ$^+$22) on CIFAR-10, ViT-Base (DBKea21) on CIFAR-100, followed by post-training low-rank truncation (AZW); further, we fine-tune BERT-Large (DCLT19) on GLUE benchmark tasks (without truncation).

## 5.1 Low-Rank Pre-Training

We compare the accuracy of models trained by Q3R against baselines LoRITa (AZW), LoRA (HSW$^+$22a), and a model trained without low-rank regularization, after post-training truncation. After training, we truncate each layer weight matrix $\mathbf{W}_k \in \Theta$ using a truncated SVD to obtain factor matrices $\mathbf{A}, \mathbf{B}$ with inner dimension $r$ and $\mathbf{W}_k \approx \mathbf{AB}$ for a range of ranks $r$ corresponding to different parameter retention percentages $p$. Depending on the experiment, we apply low-rank regularization to different subsets of weights $\{\mathbf{W}_k\}$.

**ViT-Tiny Trained on CIFAR-10.** We train ViT-Tiny on CIFAR-10 for 100 epochs using a learning rate of $\alpha = 0.00004$. We enable low-rank training for all Transformer blocks, accounting for 96% of ViT-Tiny's parameters. We conduct a hyperparameter sweep across different configurations, and Figure 1a shows the best performance achieved by each training method when rank regularization is applied to the MLP and QKV weights. From Table 1, we find that AdamQ3R retains 42.4% of the original parameters with only a 1.22% performance drop, and retains 23.2% of parameters with a 4.4% performance drop, while LoRITa consistently underperforms in comparison. As shown in Figure 1b, despite various hyperparameter configurations $\lambda$ and $d$ for LoRITa, AdamQ3R consistently outperforms LoRITa upon truncation.

**ViT-Base Trained on CIFAR-100.** To demonstrate the performance of the low-rank pre-training methods on a more challenging dataset and larger model, we train the $86M$ parameter ViT-Base from scratch for 100 epochs on CIFAR-100 with data augmentation and $\alpha = 0.0001$ (SKZ$^+$22). In line with practice for large-scale Transformers (LFX$^+$24), we apply low-rank training techniques solely to

| Model\Parameter Retention $p$ | 5% | 10% | 15% | 20% | 30% | 40% | 100% |
|---|---|---|---|---|---|---|---|
| Vanilla ViT-T | 0.1475 | 0.1252 | 0.1350 | 0.1213 | 0.1624 | 0.1524 | 0.6840 |
| LoRA | - | 0.3546 | 0.3655 | 0.3576 | - | - | - |
| LoRITa D=2, $\lambda=10^{-1}$ | 0.0989 | 0.1433 | 0.1543 | 0.2125 | 0.3247 | 0.4523 | 0.7142 |
| LoRITa D=3, $\lambda=10^{-1}$ | 0.2258 | 0.3861 | 0.4466 | 0.5035 | 0.6368 | 0.6740 | 0.7273 |
| LoRITa D=3, $\lambda=10^{-3}$ | 0.1338 | 0.2136 | 0.3839 | 0.4560 | 0.5973 | 0.6253 | **0.7449** |
| Q3R, $r_{\text{target}}=10\%$, $\lambda=10^{-3}$ | **0.2322** | 0.4085 | 0.5606 | 0.6295 | 0.6526 | 0.6654 | 0.6843 |
| Q3R, $r_{\text{target}}=15\%$, $\lambda=10^{-3}$ | 0.1796 | **0.4758** | 0.5883 | 0.6215 | 0.6455 | 0.6555 | 0.6737 |
| Q3R, $r_{\text{target}}=20\%$, $\lambda=10^{-3}$ | 0.1998 | 0.4737 | **0.6175** | 0.6511 | **0.6749** | **0.6833** | 0.6990 |
| Q3R, $r_{\text{target}}=10\%$, $\lambda=10^{-2}$ | 0.2041 | 0.4387 | 0.6115 | 0.6449 | 0.6707 | 0.6771 | 0.6889 |
| Q3R, $r_{\text{target}}=15\%$, $\lambda=10^{-2}$ | 0.1313 | 0.3896 | 0.6158 | **0.6550** | 0.6689 | 0.6801 | 0.6982 |
| Q3R, $r_{\text{target}}=20\%$, $\lambda=10^{-2}$ | 0.1870 | 0.4335 | 0.6123 | 0.6496 | 0.6744 | 0.6868 | 0.6962 |

Table 1: MLP truncation performance of ViT-T, rank regularization is applied to *both* attention (QKV) blocks and MLP blocks. For LoRA, factor ranks are adaptive to $p$.

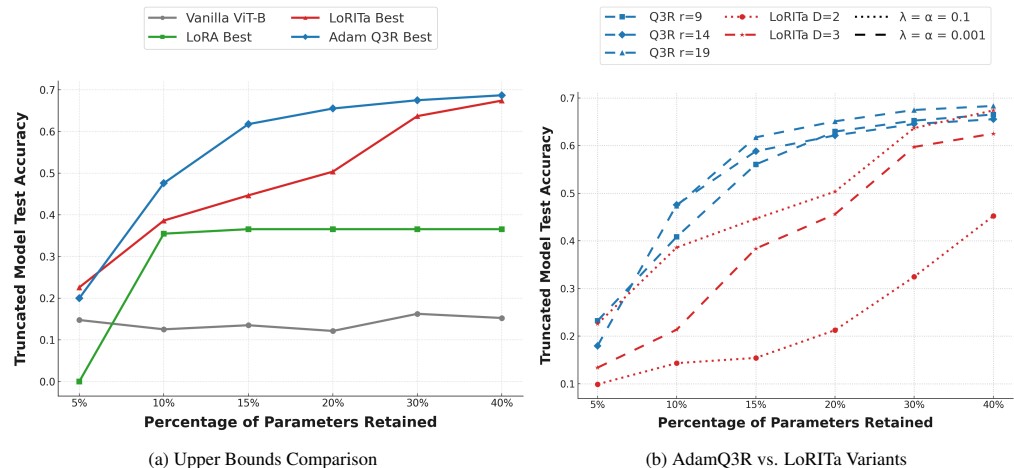

(a) Upper Bounds Comparison

(b) AdamQ3R vs. LoRITa Variants

Figure 1: Performance curves on CIFAR-10 with rank regularization applied to MLP and QKV blocks: (a) Best performance across methods, (b) AdamQ3R vs. LoRITa variants.

the multi-head self attention blocks (QKV, but not to the MLP blocks). Despite the additional size and complexity of ViT-Base compared to ViT-Tiny, Q3R remains robust and exhibits larger performance advantages with $0.40\text{-}0.44$ test accuracy at $20\%$ parameters retained, whereas LoRITa models do not exceed an accuracy of $0.25$ at the same truncation level despite their substantial overparametrization (Figures 2a and 2b).

**ViT-Tiny with Low-Rank Attention Weights.** We train ViT-Tiny for 100 epochs on CIFAR-10 (Kri09) from scratch with learning rate $\alpha = 0.0004$, with low-rank regularization applied only to attention weights. We evaluate the methods for a larger set of hyperparameters as shown in Figure 3b using layer-wise truncation levels with retained parameter percentages $p \in \{5\%, 10\%, 15\%, 20\%, 30\%, 40\%, 50\%, 60\%, 70\%, 80\%, 90\%\}$, and present results in Figure 3. Figure 3a shows that Q3R experiences almost no performance drop up to $p = 30\%$ for most parameter choices, exceeding the performance of reference methods. Figure 3b shows that the worst performing Q3R model still outperforms any LoRA, LoRITa, or vanilla ViT-Tiny below $p = 60\%$, showcasing the method's robustness.

**ViT-Base on ImageNet-1k.** We train ViT-Base on ImageNet-1k using Automatic Mixed Precision (MNA[+]18) with AutoAugmentation (CZM[+]19) for 100 epochs. Training is conducted with a learning rate of $\alpha = 4 \times 10^{-5}$, a batch size of 384, and gradient clipping (ZHSJ20) across 4 L40S GPUs. We observe that Q3R consistently outperforms the baseline model while utilizing fewer

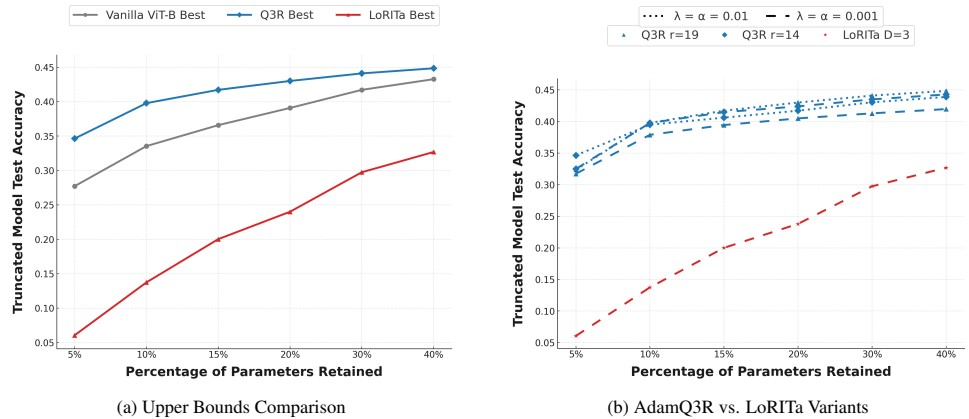

(a) Upper Bounds Comparison

(b) AdamQ3R vs. LoRITa Variants

Figure 2: Performance curves on CIFAR-100 with rank regularization applied to QKV blocks: (a) Best performance across methods, (b) AdamQ3R vs. LoRITa variants.

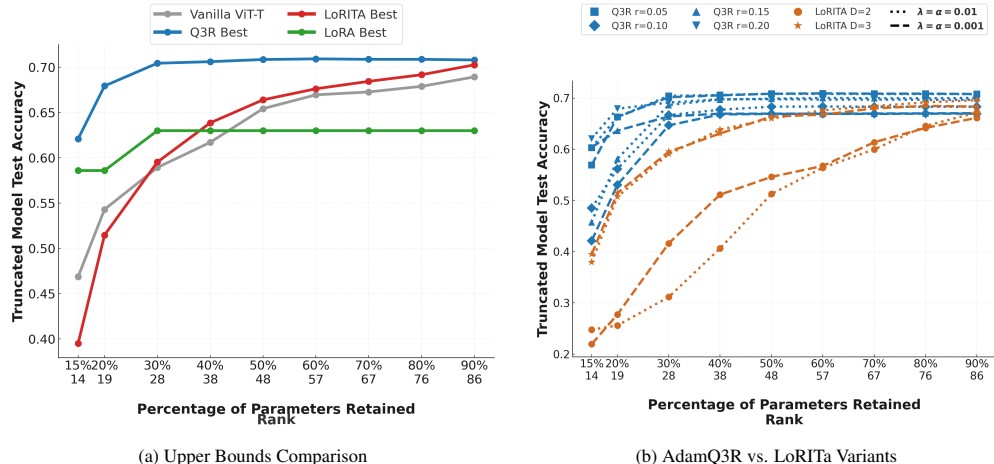

(a) Upper Bounds Comparison

(b) AdamQ3R vs. LoRITa Variants

Figure 3: Performance curves on CIFAR-10 with rank regularization applied to QKV blocks: (a) Best performance across methods, (b) AdamQ3R vs. LoRITa variants.

parameters. This performance advantage holds under two truncation paradigms: attention matrices only, and entire Transformer blocks. In both cases, Q3R maintains performance comparable to the full baseline model, as seen in Table 2.

| Optimizer \| Transformer Modules | 0.1 | 0.15 | 0.2 | 0.3 | 0.4 | 0.5 | 1 |
|---|---|---|---|---|---|---|---|
| AdamQ3R \| QKV, MLP | 0.0138 | **0.1439** | **0.3376** | 0.421 | **0.4458** | **0.4556** | **0.5816** |
| Adam \| QKV, MLP | **0.0193** | 0.0976 | 0.2950 | 0.399 | 0.4311 | 0.4523 | 0.5179 |
| AdamQ3R \| QKV | 0.1713 | 0.3016 | **0.4623** | **0.4895** | **0.4952** | **0.4975** | **0.5816** |
| Adam \| QKV | **0.2552** | **0.3366** | 0.4551 | 0.4882 | 0.4882 | 0.4937 | 0.5179 |

Table 2: ViT-Base on ImageNet-1k validation accuracy post-truncation on the last epoch

## 5.2 Low-Rank Fine-Tuning

Q3R not only induces a low-rank structure during pre-training in a memory-efficient manner, but also extends naturally to compact fine-tuning. We fine-tune pre-trained RoBERTa models on the GLUE benchmark using AdamQ3R with the proposed Q3R regularizer, and compare against full fine-tuning and LoRA (HSW$^+$22b). We impose Q3R on the weight matrices that are added to the full-rank pretrained weight matrices. For LoRA, we adopt the hyperparameters from (HSW$^+$22b), and for Q3R we cross-validate the learning rate and regularization hyperparameter $\lambda$. As shown

Table 3: GLUE Benchmark Scores

| Method | MRPC | RTE | CoLA | STS-B | SST-2 | QQP | MNLI | QNLI | Average |
|---|---|---|---|---|---|---|---|---|---|
| Dense Fine-tuning | 91.9 | **77.62** | 62.3 | **90.19** | 94.04 | 90.2 | **87.3** | 91.49 | **85.88** |
| LoRA rank=4 | 89.04 | 73.55 | 56.25 | 89.86 | **94.3** | 90.11 | 87.00 | **92.5** | 84.58 |
| Q3R rank=4 | **92.24** | 77.23 | **63.50** | 90.19 | 92.2 | **91.60** | 87.2 | 90.20 | 85.86 |

in Table 3, Q3R matches or exceeds LoRA's accuracy on most tasks and exhibits a performance closer to dense fine-tuning. These results demonstrate that Q3R can serve as a unified, low-rank training strategy—both for pre-training and fine-tuning of Transformer models. We discuss additional fine-tuning experiments in Appendix D.2.

# 6 Limitations

While our experiments showcase a robust post-truncation accuracy of Q3R-trained Transformers on vision and natural language tasks in small-to-medium scale settings that exceeds (or in the case of fine-tuning, matches) the one of other relevant low-rank training paradigms, the viability of Q3R is yet to be established across diverse architectures and large-scale problems. Fundamentally, Q3R relies on a suitable choice of the regularization strength hyper parameter $\lambda$, as well as on a suitable choice of the target rank $r_{\text{target}}$. We provide ablations about these values in Appendix E.1. While Q3R exhibits vulnerability to elevated values of $\lambda$ due to a convergence to a trivial, very low-rank matrix, this is easily detectable by monitoring the tail ratio $T(X, r) = \frac{\sum_{i=1}^{r} \sigma_i^2}{\|X\|_F^2}$ on models. In practice, we have observed stable behavior within the range $\lambda \in [0.001, 0.01]$. The target rank $r_{\text{target}}$ remains insensitive to underestimation because of the direct computation of epsilon resulting in a large $\epsilon$, and due to the monotonicity of the smoothing parameter update function (7), $\epsilon$ remains within a reasonable bound. We note that for weight matrices and iterations with large $\epsilon$, the effect of AdamQ3R resembles the one of AdamW with weight decay parameter $\lambda$ (see also (6)).

Arguably, a limitation of this work is also the fact that while the final weights after training are (for appropriate parameters) low-rank, AdamQ3R still handles *dense* weight matrix variables throughout training, which does not allow a reduction of the parameter budget *during* training, unlike recent work (MHP25). More elaborate post-training postprocessing (e.g., inspired by (WAUc+23)) might lead to further performance improvements.

# 7 Conclusion

We introduced *Quadratic Reweighted Rank Regularization* (Q3R), a principled, optimizer-compatible framework for inducing low-rank structure in deep neural network weights through explicit, continuous regularization. By majorizing a smoothed log-determinant surrogate with a quadratic model embedded in the AdamQ3R optimizer, Q3R trains weight matrices to achieve target ranks with minimal accuracy loss. This enables model compression with negligible performance degradation under reasonable parameter reductions, decreasing deployment costs and increasing throughput. Our experimental results demonstrate that Q3R generalizes across modalities and training regimes, with its design being particularly suitable for low-rank pre-training. Reducing Q3R's computational overhead, for example via low-rank subspace projections, remains to future work.

# Acknowledgement

We would like to thank Tonmoy Hasan and Arkaprava Sinha for their assistance in setting up the LLM experiments. I.G. and C.K acknowledge the support of the NSF Grant CCF-2549926 for this work.

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

## Supplementary material for `Q3R`: *Quadratic Reweighted Rank Regularizer for Effective Low-Rank Training*

In this supplementary material, we first provide theoretical justifications of the relationship between `Q3R` and the smoothed objective, expanding on Section 4.1, in Appendix A. The derivation of a `Q3R`value evaluation algorithm is provided in Appendix B. The expression used in AdamQ3R is derived in Appendix C. In Appendix D, we discuss more experimental results in both pre-training and fine-tuning, and we discuss the computational aspects. In the concluding part of this supplementary material, in Appendix E, we demonstrate the robustness of `Q3R` to hyperparameter variation.

## A   Relationship between Smoothed Log-Determinant and `Q3R`

In this section, we expand on the relationship between the $\epsilon$-smoothed log-determinant surrogate objective $F_\epsilon(\cdot)$ defined in (1). Part of this material is covered in (KM23, Section B.2) in a different context.

### A.1   Properties of Smoothed Log-Determinant

We focus first on some basic properties of the $\epsilon$-smoothed log-determinant $F_\epsilon : \mathbb{R}^{d_1 \times d_2} \to \mathbb{R}$, which, as we recall from (1), was defined for any $\mathbf{W} \in \mathbb{R}^{d_1 \times d_2}$ as

$$F_\epsilon(\mathbf{W}) := \sum_{i=1}^d f_\epsilon(\sigma_i(\mathbf{W})), \text{ where } f_\epsilon(\sigma) = \begin{cases} \epsilon^2 \left(\log(\sigma) - \log(\epsilon)\right) + \frac{1}{2}\epsilon^2, & \text{if } \sigma \geqslant \epsilon, \\ \frac{1}{2}\sigma^2, & \text{if } \sigma < \epsilon, \end{cases}$$

given $\epsilon > 0$.

As seen by its definition, $F_\epsilon(\cdot)$ is a *spectral function*, i.e., it only depends on the singular values $\sigma_1(\mathbf{W}), \sigma_2(\mathbf{W}), \ldots$ of $\mathbf{W}$, but not on any singular vector information.

Let now $d := \min(d_1, d_2)$. More precisely, we can define, following (Lew95; Bec17; LS05), a *spectral function* $F : \mathbb{R}^{d_1 \times d_2} \to \mathbb{R}$ as a function for which there exists a function $f : \mathbb{R}^d \to \mathbb{R}$ for

which $F = f \circ \sigma$, where $\sigma : \mathbb{R}^{d_1 \times d_2} \to \mathbb{R}^d$, $\mathbf{W} \mapsto \sigma(\mathbf{W}) = (\sigma_1(\mathbf{W}), \dots, \sigma_d(\mathbf{W}))$ is the function mapping matrices in $\mathbb{R}^{d_1 \times d_2}$ to its singular value vector $\sigma(\mathbf{W})$. A key towards understanding the derivative structure is that we can obtain an explicit formula for the gradient $\nabla F(\mathbf{W})$ of $F_\epsilon$ at $\mathbf{W}$ if the function $f$ in the spectral function definition is absolutely (permutation) symmetric (Bec17, Section 7.3) according to Definition A.1. It is easy to check that $f_\epsilon$ from the definition of the $\epsilon$-smoothed log-determinant $F_\epsilon(\cdot)$ satisfies this definition.

**Definition A.1** (Absolutely permutation symmetric functions). *1. Let $\mathbf{x} \in \mathbb{R}^d$. We call $r(\mathbf{x}) \in \mathbb{R}^d$ the* non-increasing rearrangement *of $\mathbf{x}$ if it holds that*

$$r(\mathbf{x})_1 \geqslant r(\mathbf{x})_2 \geqslant \dots \geqslant r(\mathbf{x})_d$$

*and there is a permutation matrix $\mathbf{P} \in \mathbb{P}^d$ such that $r(\mathbf{x})_i = (\mathbf{P}\mathbf{x})_i$ for all $i \in [d]$.*

*2. We say that a function $f : \mathbb{R}^d \to \mathbb{R}$ is* absolutely permutation symmetric *if*

$$f(\mathbf{x}) = f(r(|\mathbf{x}|)) \tag{8}$$

*for any $\mathbf{x} \in \mathbb{R}^d$.*

For ease of notation, given a vector $\mathbf{v} \in \mathbb{R}^d$, we define $\mathrm{dg}(\mathbf{v}) \in \mathbb{R}^{d_1 \times d_2}$ be the rectangular diagonal matrix such that for $v \in \mathbb{R}^d$ and any $i \in \{1, \dots, d_1\}$, $j \in \{1, \dots, d_2\}$,

$$\mathrm{dg}(\mathbf{v})_{ij} = \begin{cases} \mathbf{v}_i, & \text{if } i = j, \\ 0, & \text{else.} \end{cases}$$

Next, we cite a key result about the differentiability of spectral functions which is due to

**Proposition A.1** (Differentiability of Spectral Functions (LS05, Section 7)). *Let $F : \mathbb{R}^{d_1 \times d_2} \to \mathbb{R}$ be a spectral function $F = f \circ \sigma$ with an associated function $f : \mathbb{R}^d \to \mathbb{R}$ that is absolutely permutation symmetric. Then, $F$ is differentiable at $\mathbf{W} \in \mathbb{R}^{d_1 \times d_2}$ if and only if $f$ is differentiable at $\sigma(\mathbf{W}) \in \mathbb{R}^d$.*

*In this case, the gradient $\nabla F$ of $F$ at $\mathbf{W}$ is given by*

$$\nabla F(\mathbf{W}) = \mathbf{U} \, \mathrm{dg} \left( \nabla f(\sigma(\mathbf{W})) \right) \mathbf{V}^\top$$

*if $\mathbf{W} = \mathbf{U} \, \mathrm{dg} \left( \sigma(\mathbf{W}) \right) \mathbf{V}^\top$ is a singular value decomposition of $\mathbf{W}$ with orthogonal matrices $\mathbf{U} \in \mathbb{R}^{d_1 \times d_1}$ and $\mathbf{V} \in \mathbb{R}^{d_2 \times d_2}$.*

Using Proposition A.1, we can characterize the derivative of the $F_\epsilon$ for arbitrary $\epsilon > 0$, as established in the following lemma.

**Lemma A.2.** *Let $\epsilon > 0$ and $F_\epsilon : \mathbb{R}^{d_1 \times d_2} \to \mathbb{R}$ be the $\epsilon$-smoothed log-determinant of Equation* (1). *Then $F_\epsilon$ is differentiable with 1-Lipschitz gradient $\nabla F_\epsilon : \mathbb{R}^{d_1 \times d_2} \to \mathbb{R}^{d_1 \times d_2}$ that is given by*

$$\nabla F_\epsilon(\mathbf{W}) = \mathbf{U}_\mathbf{W} \, \mathrm{dg} \left( \frac{\sigma_i(\mathbf{W})}{\max(\sigma_i(\mathbf{W})/\epsilon, 1)^2} \right)_{i=1}^d \mathbf{V}_\mathbf{W}^\top \tag{9}$$

*for any matrix $\mathbf{W}$ with singular value decomposition $\mathbf{W} = \mathbf{U}_\mathbf{W} \, \mathrm{dg} \left( \sigma(\mathbf{W}) \right) \mathbf{V}_\mathbf{W}^\top = \mathbf{U}_\mathbf{W} \, \mathrm{dg} \left( \sigma \right) \mathbf{V}_\mathbf{W}^\top$.*

*Proof of Lemma A.2.* For the differentiability of $F_\epsilon$, as per Proposition A.1, it is sufficient to show that the function $f((\sigma_1, \dots, \sigma_d)) = \sum_{i=1}^d f_\epsilon(\sigma_i)$ with $f_\epsilon : \mathbb{R}_{\geqslant 0} \to \mathbb{R}$ as defined in (1) is differentiable at any $(\sigma_1, \dots, \sigma_d) \in \mathbb{R}_{\geqslant 0}^d$. Due to the sum structure of $f$, this will follow if $f_\epsilon$ is itself differentiable at any $\sigma \geqslant 0$.

To this, we observe that for any $\sigma > 0, \sigma \neq \epsilon$, we have that $f_\epsilon$ is differentiable at $\sigma$ with derivative

$$f_\epsilon'(\sigma) = \begin{cases} \frac{\epsilon^2}{\sigma}, & \text{if } \sigma > \epsilon \\ \sigma, & \text{if } 0 \leqslant \sigma < \epsilon. \end{cases}$$

Since $\lim_{\sigma \nearrow \epsilon} f_\epsilon'(\sigma) = \epsilon = \lim_{\sigma \searrow \epsilon} f_\epsilon'(\sigma)$, it follows that $f_\epsilon$ is also differentiable at $\sigma = \epsilon$ with $f_\epsilon'(\epsilon) = \epsilon$, and thus, differentiable on the entirety of its domain. The formula (9) follows then directly from Proposition A.1. $\qquad\square$

*Remark* A.3. It is well-known that the optimization of *convex* functions (BV04) is from many perspectives less challenging than the optimization of non-convex functions. In Section 4.1, we have claimed that the $\epsilon$-smoothed log-determinant surrogate is *not* convex. This can indeed by shown directly by invoking (LS05, Proposition 6.1), which states that a spectral function $F = f \circ \sigma$ is convex if and only if $f$ is convex. Indeed, it is easy to see that $f_\epsilon(\cdot)$ is not convex due to its logarithmic dependence on the input for large inputs, which shows that $F_\epsilon(\cdot)$ is not a convex function.

As mentioned in Section 4.1, we see from (9) that computing $\nabla F_\epsilon(\mathbf{W})$ given the matrix $\mathbf{W} \in \mathbb{R}^{d_1 \times d_2}$ indeed would require a *full* singular value decomposition that includes at least $d$ leading singular values. Defining $r(\mathbf{W}, \epsilon) := |\{i \in \{1, \ldots, d\} : \sigma_i(\mathbf{W}) > \epsilon\}|$ as in (3) and $\Sigma = \mathrm{diag}(\sigma_i(\mathbf{W}))_{i=1}^{r(\mathbf{W},\epsilon)} \in \mathbb{R}^{r(\mathbf{W},\epsilon) \times r(\mathbf{W},\epsilon)}$ as in (4), we obtain for $\mathbf{U}$ and $\mathbf{V}$ defined from the $r(\mathbf{W}, \epsilon)$ leading columns of $\mathbf{U_W} = [\mathbf{U} \quad \mathbf{U_\perp}] \in \mathbb{R}^{d_1 \times d_1}$ and $\mathbf{V_W} = [\mathbf{V} \quad \mathbf{V_\perp}] \in \mathbb{R}^{d_2 \times d_2}$ that
$$\nabla F_\epsilon(\mathbf{W}) = \epsilon^2 \mathbf{U}\Sigma^{-1}\mathbf{V}^\top + \mathbf{U_\perp}\Sigma_\perp \mathbf{V_\perp}^\top, \tag{10}$$
inserting the formula from (9), with the notation that $\Sigma_\perp = \mathrm{dg}\,(\sigma_i(\mathbf{W}))_{i=r(\mathbf{W},\epsilon)+1}^{d}$. Fundamentally, this is the key reason why a direct inclusion of the smoothed log-determinant objective into a gradient-based optimization algorithm is computationally inefficient.

Finally, we conclude with the observation that $F_\epsilon(\cdot)$ becomes *convex* if $\epsilon \gg 0$ is chosen large enough. In particular, it holds for any $\mathbf{W} \in \mathbb{R}^{d_1 \times d_2}$ that
$$F_\epsilon(\mathbf{W}) = \sum_{i=1}^{d} f_\epsilon(\sigma_i(\mathbf{W})) = \sum_{i=1}^{d} \frac{1}{2}\sigma_i^2(\mathbf{W}) = \frac{1}{2}\|\mathbf{W}\|_F^2$$
if additionally the largest singular value $\sigma_1(\mathbf{W})$ of $\mathbf{W}$ satisfies $\sigma_1(\mathbf{W}) \leqslant \epsilon$. Here, we used in the last equality that the Frobenius norm of a matrix is the $\ell_2$-norm if its singular values.

## A.2 The Quadratic Model Function Underlying Q3R

We proceed by justifying the claims made in Section 4.1 about the relationship between the $\epsilon$-smoothed log-determinant $F_\epsilon(\cdot)$, the Q3R-regularizer $\mathrm{Q3R}_{\mathbf{W}',\epsilon}(\cdot)$, the quadratic model $Q_\epsilon(\cdot \mid \mathbf{W}')$ of (2) and the reweighting operator $\mathcal{R}_{\mathbf{W}',\epsilon}(\cdot)$. To this end, we show the first statements of Lemma A.2 that characterize the reweighting operator $\mathcal{R}_{\mathbf{W}',\epsilon}(\cdot)$.

*Proof of Lemma 4.1.1.* Let $\mathbf{W}' \in \mathbb{R}^{d_1 \times d_2}$ be arbitrary with singular value decomposition $\mathbf{W}' = \mathbf{U}' \mathrm{dg}(\sigma_i(\mathbf{W}'))\mathbf{V}'^\top$. Recall from Definition 4.1 that
$$\mathcal{R}_{\mathbf{W}',\epsilon}(\mathbf{W}) = \mathbf{U}'\Sigma_{\epsilon,d_1}^{-1}\mathbf{U}'^\top \mathbf{W}\mathbf{V}'\Sigma_{\epsilon,d_2}^{-1}\mathbf{V}'^\top,$$
where $\Sigma_{\epsilon,d} = \mathrm{diag}(\max(\sigma_i(\mathbf{W}')/\epsilon, 1))_{i=1}^{d} \in \mathbb{R}^{d \times d}$ for $d \in \{d_1, d_2\}$.

To show that $\mathcal{R}_{\mathbf{W}',\epsilon} : \mathbb{R}^{d_1 \times d_2} \to \mathbb{R}^{d_1 \times d_2}$ is a positive definite operator, we consider any $\mathbf{W} \in \mathbb{R}^{d_1 \times d_2}$ such that $\mathbf{W} \neq 0$, which implies that $\|\mathbf{W}\|_F > 0$. Defining $\mathbf{Z} := \mathbf{U}'^\top \mathbf{W}\mathbf{V}'$, we see that
$$\begin{aligned}
\langle \mathbf{W}, \mathcal{R}_{\mathbf{W}',\epsilon}(\mathbf{W})\rangle &= \mathrm{tr}\left(\mathbf{W}^\top \mathcal{R}_{\mathbf{W}',\epsilon}(\mathbf{W})\right) = \mathrm{tr}\left(\mathbf{W}^\top \mathbf{U}'\Sigma_{\epsilon,d_1}^{-1}\mathbf{U}'^\top \mathbf{W}\mathbf{V}'\Sigma_{\epsilon,d_2}^{-1}\mathbf{V}'^\top\right) \\
&= \mathrm{tr}\left(\mathbf{V}'^\top \mathbf{W}^\top \mathbf{U}'\Sigma_{\epsilon,d_1}^{-1}\mathbf{U}'^\top \mathbf{W}\mathbf{V}'\Sigma_{\epsilon,d_2}^{-1}\right) = \mathrm{tr}\left(\mathbf{Z}^\top \Sigma_{\epsilon,d_1}^{-1}\mathbf{Z}\Sigma_{\epsilon,d_2}^{-1}\right) \\
&= \mathrm{tr}\left((\Sigma_{\epsilon,d_1}^{-1}\mathbf{Z})^\top \mathbf{Z}\Sigma_{\epsilon,d_2}^{-1}\right) = \sum_{i=1}^{d_1}\sum_{j=1}^{d_2}(\Sigma_{\epsilon,d_1}^{-1}\mathbf{Z})_{ij}(\mathbf{Z}\Sigma_{\epsilon,d_2}^{-1})_{ij} \\
&= \sum_{i=1}^{d_1}\sum_{j=1}^{d_2}\widetilde{\sigma}_i\widetilde{\sigma}_j\mathbf{Z}_{ij}^2
\end{aligned}$$
with $\widetilde{\sigma}_i := \max(\sigma_i(\mathbf{W}')/\epsilon, 1)^{-1}$ for $i \in \{1, \ldots, \max(d_1, d_2)\}$, with using the cyclicity of the trace in the third equality. Since $\widetilde{\sigma}_i, \widetilde{\sigma}_j > 0$ for all $i, j$, we can establish the lower bound
$$\begin{aligned}
\langle \mathbf{W}, \mathcal{R}_{\mathbf{W}',\epsilon}(\mathbf{W})\rangle &= \sum_{i=1}^{d_1}\sum_{j=1}^{d_2}\widetilde{\sigma}_i\widetilde{\sigma}_j \geqslant \min_{i=1}^{\max(d_1,d_2)}\widetilde{\sigma}_i^2 \sum_{i=1}^{d_1}\sum_{j=1}^{d_2}\mathbf{Z}_{ij}^2 = \min_{i=1}^{\max(d_1,d_2)}\widetilde{\sigma}_i^2\|\mathbf{Z}\|_F^2 \\
&= \min_{i=1}^{\max(d_1,d_2)}\widetilde{\sigma}_i^2\|\mathbf{U}'^\top \mathbf{W}\mathbf{V}'\|_F^2 = \min_{i=1}^{\max(d_1,d_2)}\widetilde{\sigma}_i^2\|\mathbf{W}\|_F^2 > 0.
\end{aligned}$$

$$\square$$

Due to the definition of Q3R (6), an implication of this is that

$$\mathrm{Q3R}_{\mathbf{W}',\epsilon}(\mathbf{W}) = \frac{1}{2}\langle \mathbf{W}, \mathcal{R}_{\mathbf{W}',\epsilon}(\mathbf{W})\rangle \geqslant 0$$

and $\mathrm{Q3R}_{\mathbf{W}',\epsilon}(\mathbf{W}) = 0 \Leftrightarrow \mathbf{W} = 0$, i.e., the value of Q3R is always non-negative and positive for non-zero matrices.

We proceed with the proof of the second statement of Lemma 4.1, which provides an explicit formula for the reweighting operator that only requires a partial SVD of $\mathbf{W}'$.

*Proof of Lemma 4.1.2.* If $\mathbf{W}' = \mathbf{U}_{\mathbf{W}'} \mathrm{dg}(\sigma_i(\mathbf{W}'))\mathbf{V}_{\mathbf{W}'}^\top$ is a full singular value decomposition of $\mathbf{W}'$ with $\mathbf{U}_{\mathbf{W}'} = [\mathbf{U} \quad \mathbf{U}_\perp] \in \mathbb{R}^{d_1 \times d_1}$ and $\mathbf{V}_{\mathbf{W}'} = [\mathbf{V} \quad \mathbf{V}_\perp] \in \mathbb{R}^{d_2 \times d_2}$, we recall that the image of a matrix $\mathbf{W} \in \mathbb{R}^{d_1 \times d_2}$ with respect to $\mathcal{R}_{\mathbf{W}',\epsilon}$ is defined (see Definition 4.1) as

$$\mathcal{R}_{\mathbf{W}',\epsilon}(\mathbf{W}) = \mathbf{U}_{\mathbf{W}'}\Sigma_{\epsilon,d_1}^{-1}\mathbf{U}_{\mathbf{W}'}^\top \mathbf{W}\mathbf{V}_{\mathbf{W}'}\Sigma_{\epsilon,d_2}^{-1}\mathbf{V}_{\mathbf{W}'}^\top, \tag{11}$$

using the definition for $\Sigma_{\epsilon,d}$ from the proof of Lemma 4.1.1 above.

With a similar argument as made in (10), we can see that

$$\mathbf{U}_{\mathbf{W}'}\Sigma_{\epsilon,d_1}^{-1}\mathbf{U}_{\mathbf{W}'}^\top = \epsilon\mathbf{U}\Sigma^{-1}\mathbf{U}^\top + \mathbf{U}_\perp\mathbf{U}_\perp^\top = \epsilon\mathbf{U}\Sigma^{-1}\mathbf{U}^\top + \mathbf{I} - \mathbf{U}\mathbf{U}^\top$$

with $\Sigma = \mathrm{diag}(\sigma_i(\mathbf{W}'))_{i=1}^{r(\mathbf{W}',\epsilon)} \in \mathbb{R}^{r(\mathbf{W}',\epsilon)\times r(\mathbf{W}',\epsilon)}$, $\mathbf{U} \in \mathbb{R}^{d_1 \times r(\mathbf{W}',\epsilon)}$ and the identity matrix $\mathbf{I}$. In the last equation, we used that $\mathbf{U}_\perp\mathbf{U}_\perp^\top$ is the projection operator onto the subspace that is *orthogonal* to the one spanned by the columns of $\mathbf{U}$. Analogously, we obtain that

$$\mathbf{V}_{\mathbf{W}'}\Sigma_{\epsilon,d_2}^{-1}\mathbf{V}_{\mathbf{W}'}^\top = \epsilon\mathbf{V}\Sigma^{-1}\mathbf{V}^\top + \mathbf{V}_\perp\mathbf{V}_\perp^\top = \epsilon\mathbf{V}\Sigma^{-1}\mathbf{V}^\top + \mathbf{I} - \mathbf{V}\mathbf{V}^\top,$$

where $\mathbf{V} \in \mathbb{R}^{d_2 \times r(\mathbf{W}',\epsilon)}$. Inserting these two equations into (11), we obtain

$$\begin{aligned}
\mathcal{R}_{\mathbf{W}',\epsilon}(\mathbf{W}) &= \mathbf{U}_{\mathbf{W}'}\Sigma_{\epsilon,d_1}^{-1}\mathbf{U}_{\mathbf{W}'}^\top \mathbf{W}\mathbf{V}_{\mathbf{W}'}\Sigma_{\epsilon,d_2}^{-1}\mathbf{V}_{\mathbf{W}'}^\top = \\
&= \left(\epsilon\mathbf{U}\Sigma^{-1}\mathbf{U}^\top + \mathbf{I} - \mathbf{U}\mathbf{U}^\top\right)\mathbf{W}\left(\epsilon\mathbf{V}\Sigma^{-1}\mathbf{V}^\top + \mathbf{I} - \mathbf{V}\mathbf{V}^\top\right) \\
&= \epsilon^2\mathbf{U}\Sigma^{-1}\mathbf{U}^\top\mathbf{W}\mathbf{V}\Sigma^{-1}\mathbf{V}^\top + \epsilon\mathbf{U}\Sigma^{-1}\mathbf{U}^\top\mathbf{W}\left(\mathbf{I} - \mathbf{V}\mathbf{V}^\top\right) \\
&\quad + \epsilon\left(\mathbf{I} - \mathbf{U}\mathbf{U}^\top\right)\mathbf{W}\mathbf{V}\Sigma^{-1}\mathbf{V}^\top + \left(\mathbf{I} - \mathbf{U}\mathbf{U}^\top\right)\mathbf{W}\left(\mathbf{I} - \mathbf{V}\mathbf{V}^\top\right),
\end{aligned}$$

where the last equality shows the statement of Lemma 4.1.2. $\square$

As a preparation for the proof of the last statement of Lemma 4.1, we formulate the following lemma which relates the gradient of $F_\epsilon$ at $\mathbf{W}$ with the reweighting operator.

**Lemma A.4** (Gradient Condition). *Let $\epsilon > 0$ For any $\mathbf{W} \in \mathbb{R}^{d_1 \times d_2}$, the reweighting operator $\mathcal{R}_{\mathbf{W},\epsilon} : \mathbb{R}^{d_1 \times d_2} \to \mathbb{R}^{d_1 \times d_2}$ satisfies*

$$\mathcal{R}_{\mathbf{W},\epsilon}(\mathbf{W}) = \nabla F_\epsilon(\mathbf{W}), \tag{12}$$

*where $\nabla F_\epsilon(\mathbf{W})$ is the gradient of the $\epsilon$-smoothed log-determinant at $\mathbf{W}$.*

*Proof.* If $\mathbf{W} = \mathbf{U}_{\mathbf{W}} \mathrm{dg}(\sigma_i(\mathbf{W}))\mathbf{V}_{\mathbf{W}}^\top$ is a singular value decomposition of $\mathbf{W}$ with $\mathbf{U}_{\mathbf{W}} = [\mathbf{U} \quad \mathbf{U}_\perp] \in \mathbb{R}^{d_1 \times d_1}$ and $\mathbf{V}_{\mathbf{W}} = [\mathbf{V} \quad \mathbf{V}_\perp] \in \mathbb{R}^{d_2 \times d_2}$, we observe that

$$\begin{aligned}
\mathcal{R}_{\mathbf{W},\epsilon}(\mathbf{W}) &= \mathbf{U}_{\mathbf{W}}\Sigma_{\epsilon,d_1}^{-1}\mathbf{U}_{\mathbf{W}}^\top \left(\mathbf{U}_{\mathbf{W}} \mathrm{dg}(\sigma_i(\mathbf{W}))\mathbf{V}_{\mathbf{W}}^\top\right)\mathbf{V}_{\mathbf{W}}\Sigma_{\epsilon,d_2}^{-1}\mathbf{V}_{\mathbf{W}}^\top \\
&= \mathbf{U}_{\mathbf{W}}\Sigma_{\epsilon,d_1}^{-1}\mathrm{dg}(\sigma_i(\mathbf{W}))\Sigma_{\epsilon,d_2}^{-1}\mathbf{V}_{\mathbf{W}}^\top \\
&= \mathbf{U}_{\mathbf{W}} \mathrm{dg}\left(\frac{\sigma_i(\mathbf{W})}{\max(\sigma_i(\mathbf{W})/\epsilon,1)^2}\right)_{i=1}^d \mathbf{V}_{\mathbf{W}}^\top = \nabla F_\epsilon(\mathbf{W}),
\end{aligned}$$

using the gradient formula Lemma A.2 in the last equality. $\square$

As a corollary of Lemma A.4, we see that for $\mathbf{W}' = \mathbf{W}$, the gradient of the Q3R regularizer satisfies

$$\nabla\, \mathrm{Q3R}_{\mathbf{W},\epsilon}(\mathbf{W}) = \nabla F_\epsilon(\mathbf{W}).$$

This is this a direct implication of Lemma A.4 since

$$\nabla_{\mathbf{W}}\, \mathrm{Q3R}_{\mathbf{W}'=\mathbf{W},\epsilon}(\mathbf{W}) = \nabla_{\mathbf{W}} \left( \frac{1}{2}\langle \mathbf{W}, \mathcal{R}_{\mathbf{W}'=\mathbf{W},\epsilon}(\mathbf{W})\rangle \right) = \mathcal{R}_{\mathbf{W},\epsilon}(\mathbf{W})$$

using the self-adjointness (see, e.g., (KM23, Appendix B)) of $\mathcal{R}_{\mathbf{W},\epsilon}$.

The gradient condition (12) enables us to equate the definition of the quadratic model function (2) $Q_\epsilon(\cdot \mid \mathbf{W}')$ (which is, up to a constant that depends on $\epsilon$ and $\mathbf{W}'$ the same as the value of Q3R) with the standard quadratic model form of (5).

*Proof of Lemma 4.1.3.* Let $\mathbf{W}, \mathbf{W}' \in \mathbb{R}^{d_1 \times d_2}$ be arbitrary. To show the equation (5), we start with its right hand side. By inserting (12), we obtain

$$F_\epsilon(\mathbf{W}') + \langle \nabla F_\epsilon(\mathbf{W}'), \mathbf{W} - \mathbf{W}'\rangle + \tfrac{1}{2}\langle \mathbf{W} - \mathbf{W}', \mathcal{R}_{\mathbf{W}',\epsilon}(\mathbf{W} - \mathbf{W}')\rangle$$

$$= F_\epsilon(\mathbf{W}') + \langle \mathcal{R}_{\mathbf{W}',\epsilon}(\mathbf{W}'), \mathbf{W} - \mathbf{W}'\rangle + \frac{1}{2}\langle \mathbf{W}, \mathcal{R}_{\mathbf{W}',\epsilon}(\mathbf{W})\rangle - \langle \mathcal{R}_{\mathbf{W}',\epsilon}(\mathbf{W}'), \mathbf{W}\rangle$$

$$+ \frac{1}{2}\langle \mathbf{W}', \mathcal{R}_{\mathbf{W}',\epsilon}(\mathbf{W}')\rangle$$

$$= F_\epsilon(\mathbf{W}') + \frac{1}{2}\langle \mathbf{W}, \mathcal{R}_{\mathbf{W}',\epsilon}(\mathbf{W})\rangle - \frac{1}{2}\langle \mathbf{W}', \mathcal{R}_{\mathbf{W}',\epsilon}(\mathbf{W}')\rangle$$

$$=: Q_\epsilon(\mathbf{W} \mid \mathbf{W}'),$$

where we also use the self-adjointness of $\mathcal{R}_{\mathbf{W}',\epsilon}(\cdot)$ in the first equality. The last expression corresponds to the definition of the quadratic model $Q_\epsilon(\cdot \mid \mathbf{W}')$ of $F_\epsilon(\cdot)$ given the expansion point $\mathbf{W}'$. This concludes the proof. □

From this proof, it becomes clear that the quadratic model $Q_\epsilon(\cdot \mid \mathbf{W}')$ is a pure quadratic model with vanishing linear term. This implies that, for example, $Q_\epsilon(-\mathbf{W} \mid \mathbf{W}') = Q_\epsilon(\mathbf{W} \mid \mathbf{W}')$ for all $\mathbf{W}$, which reflects the geometry of the smoothed log-determinant $F_\epsilon(\cdot)$ better than a mixed quadratic model function as it likewise satisfies $F_\epsilon(-\mathbf{W}) = F_\epsilon(\mathbf{W})$.

# B Computation of Q3R value

In this section, we provide an implementable algorithm for evaluating the Q3R regularizer $\mathrm{Q3R}_{\mathbf{W}',\epsilon}(\mathbf{W})$ as defined in (6), defined in Algorithm 3 below.

We note that strictly speaking, evaluating $\mathrm{Q3R}_{\mathbf{W}',\epsilon}(\mathbf{W})$ is never necessary in a training scheme such as AdamQ3R; however, evaluating $\mathrm{Q3R}_{\mathbf{W}',\epsilon}(\mathbf{W})$ might be insightful to keep track of the extent of the regularization.

First, we decompose the reweighting operator image such that

$$\mathcal{R}_{\mathbf{W}',\epsilon}(\mathbf{W}) = T_1^\epsilon + T_2^\epsilon + T_3^\epsilon + T_4^\epsilon$$

with

$$T_1^\epsilon = \epsilon^2\, \mathbf{U}\,\boldsymbol{\Sigma}^{-1}\mathbf{U}^\top\,\mathbf{W}\,\mathbf{V}\,\boldsymbol{\Sigma}^{-1}\mathbf{V}^\top,$$
$$T_2^\epsilon = \epsilon\, \mathbf{U}\,\boldsymbol{\Sigma}^{-1}\mathbf{U}^\top\,\mathbf{W}\,(\mathbf{I} - \mathbf{V}\mathbf{V}^\top),$$
$$T_3^\epsilon = \epsilon\, (\mathbf{I} - \mathbf{U}\mathbf{U}^\top)\,\mathbf{W}\,\mathbf{V}\,\boldsymbol{\Sigma}^{-1}\mathbf{V}^\top,$$
$$T_4^\epsilon = (\mathbf{I} - \mathbf{U}\mathbf{U}^\top)\,\mathbf{W}\,(\mathbf{I} - \mathbf{V}\mathbf{V}^\top).$$

Defining

$$I_1 = \langle \mathbf{W}, T_1^\epsilon\rangle = \epsilon^2\, \mathrm{tr}\big(\mathbf{W}^\top\mathbf{U}\,\boldsymbol{\Sigma}^{-1}\mathbf{U}^\top\mathbf{W}\,\mathbf{V}\,\boldsymbol{\Sigma}^{-1}\mathbf{V}^\top\big),$$
$$I_2 = \langle \mathbf{W}, T_2^\epsilon\rangle = \epsilon\, \mathrm{tr}\big(\mathbf{W}^\top\mathbf{U}\,\boldsymbol{\Sigma}^{-1}\mathbf{U}^\top\mathbf{W}\,(\mathbf{I} - \mathbf{V}\mathbf{V}^\top)\big),$$
$$I_3 = \langle \mathbf{W}, T_3^\epsilon\rangle = \epsilon\, \mathrm{tr}\big(\mathbf{W}^\top(\mathbf{I} - \mathbf{U}\mathbf{U}^\top)\,\mathbf{W}\,\mathbf{V}\,\boldsymbol{\Sigma}^{-1}\mathbf{V}^\top\big),$$
$$I_4 = \langle \mathbf{W}, T_4^\epsilon\rangle = \mathrm{tr}\big(\mathbf{W}^\top(\mathbf{I} - \mathbf{U}\mathbf{U}^\top)\,\mathbf{W}\,(\mathbf{I} - \mathbf{V}\mathbf{V}^\top)\big),$$

we can write
$$\langle \mathbf{W}, \mathcal{R}_{\mathbf{W}',\epsilon}(\mathbf{W})\rangle = I_1 + I_2 + I_3 + I_4.$$

**Apply cyclicity to $I_1, I_2, I_3$.**

$$I_1 = \epsilon^2 \operatorname{tr}\big(\mathbf{V}^\top \mathbf{W}^\top \mathbf{U}\, \mathbf{\Sigma}^{-1} \mathbf{U}^\top \mathbf{W}\, \mathbf{V}\, \mathbf{\Sigma}^{-1}\big),$$
$$I_2 = \epsilon\Big[ \operatorname{tr}\big(\mathbf{W}^\top \mathbf{U}\, \mathbf{\Sigma}^{-1} \mathbf{U}^\top \mathbf{W}\big) \;-\; \operatorname{tr}\big(\mathbf{W}^\top \mathbf{U}\, \mathbf{\Sigma}^{-1} \mathbf{U}^\top \mathbf{W}\, \mathbf{V}\mathbf{V}^\top\big)\Big],$$
$$I_3 = \epsilon\Big[ \operatorname{tr}\big(\mathbf{W}^\top \mathbf{W}\, \mathbf{V}\, \mathbf{\Sigma}^{-1} \mathbf{V}^\top\big) \;-\; \operatorname{tr}\big(\mathbf{W}^\top \mathbf{U}\mathbf{U}^\top \mathbf{W}\, \mathbf{V}\, \mathbf{\Sigma}^{-1} \mathbf{V}^\top\big)\Big].$$

**Expand $I_4$.**

$$I_4 = \operatorname{tr}\big(\mathbf{W}^\top \mathbf{W}\big) \;-\; \operatorname{tr}\big(\mathbf{W}^\top \mathbf{W}\, \mathbf{V}\, \mathbf{V}^\top\big) \;-\; \operatorname{tr}\big(\mathbf{W}^\top \mathbf{U}\mathbf{U}^\top \mathbf{W}\big) \;+\; \operatorname{tr}\big(\mathbf{W}^\top \mathbf{U}\mathbf{U}^\top \mathbf{W}\, \mathbf{V}\, \mathbf{V}^\top\big).$$

**Group terms.**

$$\begin{aligned}
\langle \mathbf{W}, \mathcal{R}_{\mathbf{W}',\epsilon}(\mathbf{W})\rangle = {}& \operatorname{tr}(\mathbf{W}^\top \mathbf{W}) \\
&+ \epsilon^2 \operatorname{tr}\big(\mathbf{V}^\top \mathbf{W}^\top \mathbf{U}\, \mathbf{\Sigma}^{-1} \mathbf{U}^\top \mathbf{W}\, \mathbf{V}\, \mathbf{\Sigma}^{-1}\big) \\
&+ \epsilon \operatorname{tr}\big(\mathbf{W}^\top \mathbf{U}\, \mathbf{\Sigma}^{-1} \mathbf{U}^\top \mathbf{W}\big) \;-\; \epsilon \operatorname{tr}\big(\mathbf{W}^\top \mathbf{U}\, \mathbf{\Sigma}^{-1} \mathbf{U}^\top \mathbf{W}\, \mathbf{V}\mathbf{V}^\top\big) \\
&+ \epsilon \operatorname{tr}\big(\mathbf{W}^\top \mathbf{W}\, \mathbf{V}\, \mathbf{\Sigma}^{-1} \mathbf{V}^\top\big) \;-\; \epsilon \operatorname{tr}\big(\mathbf{W}^\top \mathbf{U}\mathbf{U}^\top \mathbf{W}\, \mathbf{V}\, \mathbf{\Sigma}^{-1} \mathbf{V}^\top\big) \\
&- \operatorname{tr}\big(\mathbf{W}^\top \mathbf{W}\, \mathbf{V}\, \mathbf{V}^\top\big) \;-\; \operatorname{tr}\big(\mathbf{W}^\top \mathbf{U}\mathbf{U}^\top \mathbf{W}\big) \;+\; \operatorname{tr}\big(\mathbf{W}^\top \mathbf{U}\mathbf{U}^\top \mathbf{W}\, \mathbf{V}\, \mathbf{V}^\top\big).
\end{aligned}$$

Then by rearranging each pair of trace-terms we arrive at

$$\begin{aligned}
\langle \mathbf{W}, \mathcal{R}_{\mathbf{W}',\epsilon}(\mathbf{W})\rangle = {}& \operatorname{tr}\big(\mathbf{W}^\top \mathbf{W}\big) \\
&+ \operatorname{tr}\big(\mathbf{U}^\top \mathbf{W}\, \mathbf{W}^\top \mathbf{U}\, (\epsilon\, \mathbf{\Sigma}^{-1} - \mathbf{I})\big) \\
&+ \operatorname{tr}\big(\mathbf{V}^\top \mathbf{W}^\top \mathbf{W}\, \mathbf{V}\, (\epsilon\, \mathbf{\Sigma}^{-1} - \mathbf{I})\big) \\
&+ \operatorname{tr}\big(\mathbf{V}^\top \mathbf{W}^\top \mathbf{U}\, (\epsilon\, \mathbf{\Sigma}^{-1} - \mathbf{I})\, \mathbf{U}^\top \mathbf{W}\, \mathbf{V}\, (\epsilon\, \mathbf{\Sigma}^{-1} - \mathbf{I})\big).
\end{aligned}$$

For each iteration we calculate the quadratic regularizer $\langle \mathbf{W}, \mathcal{R}_{\mathbf{W}',\epsilon}(\mathbf{W})\rangle$ for weight matrices $\mathbf{W}$. For algorithmic simplicity, We now re-arrange the summand of our `Q3R` regularizer and show the simplified expression for this inner product in Algorithm 3. We later show in Equation (13), where we derive the expression of this inner product and we show that this matches the one proposed in Algorithm 3.

$$f = \underbrace{\operatorname{tr}\big(\mathbf{W}^\top \mathbf{W}\big)}_{t_1} + \underbrace{\operatorname{tr}\big(\mathbf{V}^\top \mathbf{W}^\top \mathbf{U}\, \mathbf{S}\, \mathbf{U}^\top \mathbf{W}\, \mathbf{V}\, \mathbf{S}\big)}_{t_2} + \underbrace{\operatorname{tr}\big(\mathbf{V}^\top \mathbf{W}^\top \mathbf{W}\, \mathbf{V}\, \mathbf{S}\big)}_{t_3} + \underbrace{\operatorname{tr}\big(\mathbf{U}^\top \mathbf{W}\, \mathbf{W}^\top \mathbf{U}\, \mathbf{S}\big)}_{t_4},$$
$$\mathbf{S} = \epsilon\, \mathbf{\Sigma}^{-1} - \mathbf{I},$$
$$\mathbf{T} = \mathbf{W}\, \mathbf{V}, \quad \mathbf{B} = \mathbf{W}^\top \mathbf{U}, \quad \mathbf{C} = \mathbf{U}^\top \mathbf{W}\, \mathbf{V}, \quad \mathbf{M} = \mathbf{S}^{1/2}\, \mathbf{C}\, \mathbf{S}^{1/2}.$$

Using $\operatorname{tr}(\mathbf{A}^\top \mathbf{A}) = \|\mathbf{A}\|_F^2$ and cyclicity:

$$\begin{aligned}
t_1 &= \operatorname{tr}(\mathbf{W}^\top \mathbf{W}) = \|\mathbf{W}\|_F^2, \\
t_2 &= \operatorname{tr}\big(\mathbf{S}\, (\mathbf{U}^\top \mathbf{W}\mathbf{V})\, \mathbf{S}\, (\mathbf{U}^\top \mathbf{W}\mathbf{V})\big) = \|\mathbf{M}\|_F^2, \\
t_3 &= \operatorname{tr}\big(\mathbf{S}\, (\mathbf{W}\mathbf{V})^\top (\mathbf{W}\mathbf{V})\big) = \|\mathbf{T}\, \mathbf{S}^{1/2}\|_F^2, \\
t_4 &= \operatorname{tr}\big(\mathbf{S}\, (\mathbf{W}^\top \mathbf{U})^\top (\mathbf{W}^\top \mathbf{U})\big) = \|\mathbf{B}\, \mathbf{S}^{1/2}\|_F^2.
\end{aligned}$$

Putting it all together, with $\mathbf{R} = \mathbf{S}^{1/2}$:

$$f(\mathbf{W}, \mathbf{U}, \mathbf{V}) = \|\mathbf{W}\|_F^2 + \|\mathbf{M}\|_F^2 + \|\mathbf{T}\, \mathbf{R}\|_F^2 + \|\mathbf{B}\, \mathbf{R}\|_F^2, \qquad \mathbf{R} = (\epsilon\, \mathbf{\Sigma}^{-1} - \mathbf{I})^{1/2}. \tag{13}$$

**Algorithm 3** Computation of the Q3R function value $\text{Q3R}_{\mathbf{W}',\epsilon}(\mathbf{W})$

---

1: **Input:** $W \in \mathbb{R}^{d_1 \times d_2}$, $U \in \mathbb{R}^{d_1 \times r}$, $V \in \mathbb{R}^{d_2 \times r}$, $S \in \mathbb{R}^{r \times r}$
2: **Output:** e $f(W, U, V, S) = \|W\|_F^2 + \|M\|_F^2 + \|T\,R\|_F^2 + \|B\,R\|_F^2$
3: **1. Compute projections**
4: $T \leftarrow W\,V$ $\hspace{10cm}$ $O(d_1 d_2 r)$
5: $B \leftarrow W^\top U$ $\hspace{10cm}$ $O(d_1 d_2 r)$
6: **2. Form intermediate products**
7: $C \leftarrow U^\top T$ $\hspace{10cm}$ $O(d_1 r^2)$
8: Compute symmetric square-root $R$ of $S$: $R\,R^\top = S$ $\hspace{3cm}$ $O(r^3)$
9: $M \leftarrow R\,C\,R^\top$ $\hspace{10cm}$ $O(r^3)$
10: **3. Evaluate Frobenius norms**
11: $t_1 \leftarrow \|W\|_F^2$ $\hspace{10cm}$ $O(d_1 d_2)$
12: $t_2 \leftarrow \|M\|_F^2$ $\hspace{10cm}$ $O(r^2)$
13: $t_3 \leftarrow \|T\,R\|_F^2$ $\hspace{9cm}$ $O(d_1 r^2 + d_1 r)$
14: $t_4 \leftarrow \|B\,R\|_F^2$ $\hspace{9cm}$ $O(d_2 r^2 + d_2 r)$
15: **return** $f \leftarrow t_1 + t_2 + t_3 + t_4$

---

To summarize, we obtain a total FLOP count of

$$\boxed{T_{\text{total}} = 2\,d_1 d_2 r \ + \ (d_1 + d_2)r^2 \ + \ 2r^3 \ + \ \big(d_1 d_2 + (d_1 + d_2)r + r^2\big) = O\big(d_1 d_2 r \ + \ (d_1 + d_2)r^2 \ + \ r^3\big)}$$

for evaluating $\text{Q3R}_{\mathbf{W}',\epsilon}(\mathbf{W})$.

## C Computation of the Gradient of Q3R

From Equation (4), we rearrange the summand for algorithmic simplicity

$$\mathcal{R}_{\mathbf{W}',\epsilon}(\mathbf{W}) = \epsilon^2\,\mathbf{U}\,\mathbf{\Sigma}^{-1}\mathbf{U}^\top\,\mathbf{W}\,\mathbf{V}\,\mathbf{\Sigma}^{-1}\mathbf{V}^\top \ + \ \epsilon\,\mathbf{U}\,\mathbf{\Sigma}^{-1}\mathbf{U}^\top\,\mathbf{W}\,(\mathbf{I} - \mathbf{V}\mathbf{V}^\top)$$
$$+ \ \epsilon\,(\mathbf{I} - \mathbf{U}\mathbf{U}^\top)\,\mathbf{W}\,\mathbf{V}\,\mathbf{\Sigma}^{-1}\mathbf{V}^\top \ + \ (\mathbf{I} - \mathbf{U}\mathbf{U}^\top)\,\mathbf{W}\,(\mathbf{I} - \mathbf{V}\mathbf{V}^\top).$$

Using $\epsilon\,\mathbf{\Sigma}^{-1} = \mathbf{I} + \mathbf{S}$, we rewrite each term:

$$T_1^\epsilon = \epsilon^2\,\mathbf{U}\,\mathbf{\Sigma}^{-1}\mathbf{U}^\top\,\mathbf{W}\,\mathbf{V}\,\mathbf{\Sigma}^{-1}\mathbf{V}^\top = \mathbf{U}\,(\mathbf{I} + \mathbf{S})\,\mathbf{U}^\top\,\mathbf{W}\,\mathbf{V}\,(\mathbf{I} + \mathbf{S})\,\mathbf{V}^\top,$$
$$T_2^\epsilon = \epsilon\,\mathbf{U}\,\mathbf{\Sigma}^{-1}\mathbf{U}^\top\,\mathbf{W}\,(\mathbf{I} - \mathbf{V}\mathbf{V}^\top) = \mathbf{U}\,(\mathbf{I} + \mathbf{S})\,\mathbf{U}^\top\,\mathbf{W}\,(\mathbf{I} - \mathbf{V}\mathbf{V}^\top),$$
$$T_3^\epsilon = \epsilon\,(\mathbf{I} - \mathbf{U}\mathbf{U}^\top)\,\mathbf{W}\,\mathbf{V}\,\mathbf{\Sigma}^{-1}\mathbf{V}^\top = (\mathbf{I} - \mathbf{U}\mathbf{U}^\top)\,\mathbf{W}\,\mathbf{V}\,(\mathbf{I} + \mathbf{S})\,\mathbf{V}^\top,$$
$$T_4^\epsilon = (\mathbf{I} - \mathbf{U}\mathbf{U}^\top)\,\mathbf{W}\,(\mathbf{I} - \mathbf{V}\mathbf{V}^\top).$$

Collecting like-terms in powers of $\mathbf{S}$ gives the compact form

$$\boxed{\mathcal{R}_{\mathbf{W}',\epsilon}(\mathbf{W}) = \underbrace{\mathbf{W}}_{T_1} + \underbrace{\mathbf{U}\,\mathbf{S}\,\mathbf{U}^\top\,\mathbf{W}}_{T_2} + \underbrace{\mathbf{W}\,\mathbf{V}\,\mathbf{S}\,\mathbf{V}^\top}_{T_3} + \underbrace{\mathbf{U}\,\mathbf{S}\,\mathbf{U}^\top\,\mathbf{W}\,\mathbf{V}\,\mathbf{S}\,\mathbf{V}^\top}_{T_4}}$$

with $\mathbf{S} = \epsilon\,\mathbf{\Sigma}^{-1} - \mathbf{I}$.

Now, we deduce the gradient in the following algorithm that is used in line 12 of Algorithm 2. We explain the step-by-step computation of our $R_{\mathbf{W}',\epsilon}(\mathbf{W})$ stated below.

**Algorithm 4** COMPUTATION GRADIENT OF `Q3R` : Compute $\mathcal{R}_{\mathbf{W}',\epsilon}(\mathbf{W})$

---

1: **Input:** $\mathbf{W} \in \mathbb{R}^{d_1 \times d_2}$, $\mathbf{U} \in \mathbb{R}^{d_1 \times r}$, $\mathbf{V} \in \mathbb{R}^{d_2 \times r}$, singular values $\boldsymbol{\sigma} \in \mathbb{R}^r_{>0}$, threshold $\epsilon$
2: **Output:** $\mathbf{G} = \mathcal{R}_{\mathbf{W}',\epsilon}(\mathbf{W})$

---

3:
4: $s \leftarrow 1/\max(\boldsymbol{\sigma},\ 1)$          $\triangleright$ $r$ element-wise comparisons
5: $S \leftarrow \mathrm{diag}(s)$          $\triangleright$ create diagonal matrix of shape $r \times r$
6: $S_{\mathrm{shift}} \leftarrow \epsilon S - I$          $\triangleright$ $r$ subtractions on diagonal

---

7:
8: $A \leftarrow \mathbf{U}^\top \mathbf{W}$          $\triangleright$ $r \times d_1$ by $d_1 \times d_2 \rightarrow r \times d_2$ (cost: $rd_1d_2$)
9: $B \leftarrow \mathbf{W}\,\mathbf{V}$          $\triangleright$ $d_1 \times d_2$ by $d_2 \times r \rightarrow d_1 \times r$ (cost: $d_1d_2r$)

---

10:
11: $C \leftarrow A \cdot \mathbf{V}$          $\triangleright$ $r \times d_2$ by $d_2 \times r \rightarrow r \times r$ (cost: $rd_2r$)
12: $E \leftarrow S_{\mathrm{shift}} \cdot C \cdot S_{\mathrm{shift}}$          $\triangleright$ $r \times r$ triple product (cost: $2r^3$)
13: **for** $i = 1, \ldots, r$ **do**
14:      **for** $j = 1, \ldots, r$ **do**
15:          $E_{ij} \leftarrow (S_{\mathrm{shift}})_{ii} \cdot C_{ij} \cdot (S_{\mathrm{shift}})_{jj}$
16: **end for end for**          $\triangleright$ elementwise scalar products (cost: $r^2$)
17: $T_2 \leftarrow \mathbf{U} \cdot E \cdot \mathbf{V}^\top$          $\triangleright$
         • $\mathbf{U} \cdot E$: $d_1 \times r$ by $r \times r \rightarrow d_1 \times r$ (cost: $d_1r^2$)
         • Then $\cdot\mathbf{V}^\top$: $d_1 \times r$ by $r \times d_2 \rightarrow d_1 \times d_2$ (cost: $d_1d_2r$)

---

18:
19: $T_1 \leftarrow \mathbf{W}$          $\triangleright$ copy or identity operation ($d_1 \times d_2$)
20: $T_3 \leftarrow B \cdot S_{\mathrm{shift}} \cdot \mathbf{V}^\top$          $\triangleright$
         • $B \cdot S_{\mathrm{shift}}$: $d_1 \times r$ by $r \times r \rightarrow d_1 \times r$ (cost: $d_1r^2$)
         • Then $\cdot\mathbf{V}^\top$: $d_1 \times r$ by $r \times d_2 \rightarrow d_1 \times d_2$ (cost: $d_1d_2r$)
21: $D \leftarrow S_{\mathrm{shift}} \cdot A$          $\triangleright$ $r \times r$ by $r \times d_2 \rightarrow r \times d_2$ (cost: $r^2d_2$)
22: $T_4 \leftarrow \mathbf{U} \cdot D$          $\triangleright$ $d_1 \times r$ by $r \times d_2 \rightarrow d_1 \times d_2$ (cost: $d_1rd_2$)

---

23:
24: **gradient** $\leftarrow T_1 + T_2 + T_3 + T_4$          $\triangleright$ elementwise addition of $d_1 \times d_2$ matrices
25: **return gradient**

---

# D    Experimental Results

In this section, we provide details regarding the experimental training methodology of Section 5 as well as some additional data of these experiments.

## D.1    Experimental Protocol

In the experiments, we compared the training of unregularized models, as well as models regularized by `Q3R` , LoRITa (AZW) and LoRA (HSW$^+$22a). Due to the limitations of some techniques not providing strong truncation guide lines, we truncate each method at various truncation ranks $r$. Ensuring such that $r$ is chosen to always ensure that the factor matrix pairs always have less original parameters, to minimize a practical environment. In Table 1 we find that running with $p = .20$ or $r = 19$ and $\lambda = 0.001$ performs best given the truncation and accuracy trade-offs.

**ViT Hyperparameter Selection** We selected each model's learning rate based on the performance of the unmodified ViT model for each respective dataset. All ViT models were configured with an input resolution of 224×224 pixels and utilized patch size 16 for tokenization. LoRITa hyperparameter optimization was conducted through grid search with regularization parameter $\lambda \in \{10^{-1}, 10^{-2}, 10^{-3}, 10^{-4}\}$ and rank parameter $d \in \{1, 2, 3\}$ to ensure optimal performance. The best-performing configuration from three independent runs was selected for evaluation across additional datasets. LoRA was evaluated across various target ranks, with the proportion parameter $p$ selected from $p \in \{0.05, 0.15, 0.3, 0.4, 0.5, 0.6, 0.7, 0.8, 0.9\}$. AdamQ3R underwent grid search optimization with regularization parameter $\lambda \in \{10^{-3}, 10^{-2}, 10^{-1}\}$ and proportion parameter $r \in \{0.05, 0.1, 0.15, 0.2\}$, with a stable $period = 5$. The regularization strength remained constant regardless of matrix dimensions. The QKV projection matrices were selected as the target

for modification due to their prevalence in the literature as candidates for model compression and adaptation.

**CIFAR-100 Data Augmentation**    For CIFAR-100 experiments, we applied comprehensive data augmentation during training, including random cropping with 4-pixel padding from the original 32×32 images, random horizontal flipping, and subsequent resizing to 224×224 pixels to match the ViT input requirements. Images were normalized using channel-wise means of (0.4914, 0.4822, 0.4465) and standard deviations of (0.2023, 0.1994, 0.2010), corresponding to the CIFAR-100 dataset statistics. The test set underwent only resizing to 224×224 pixels and the same normalization procedure, without augmentation.

**Fine-Tuning Experimental Details**    We fine-tuned the pre-trained RoBERTa-Base model (from Hugging Face) on all nine GLUE tasks using a maximum sequence length of 512 and a batch size of 16 (30 for CoLA). We run all fine-tuning experiments with a target rank as low as $4$. The best performing setup had 'reweighting period' of 3 and $\lambda \in \{1.5, 2\}$. As shown in Table 3, we compare `Q3R` with LoRA and dense fine-tuning. The tasks and their metrics are summarized below:

- **Single-Sentence Classification**
    - CoLA: 8.5 k train / 1 k test; linguistic acceptability; Matthews correlation
    - SST-2: 67 k train / 1.8 k test; sentiment classification; accuracy
- **Similarity & Paraphrase**
    - MRPC: 3.7 k train / 1.7 k test; paraphrase detection; accuracy / F1
    - STS-B: 7 k train / 1.4 k test; sentence similarity; Pearson / Spearman correlation
    - QQP: 364 k train / 391 k test; question paraphrase detection; accuracy / F1
- **Natural Language Inference**
    - MNLI: 393 k train / 20 k matched + 20 k mismatched test; entailment classification; accuracy
    - QNLI: 105 k train / 5.4 k test; question–answer entailment; accuracy
    - RTE: 2.5 k train / 3 k test; textual entailment; accuracy
    - WNLI: 634 train / 146 test; coreference-based inference; accuracy

### D.2    Fine-tuning Experiments

To assess the potential of `Q3R` for fine-tuning large language models, we conducted an additional experiment on Llama 3.2–3B using an NVIDIA A5000 GPU. Since the original `meta-llama/Llama-3.2-3B` checkpoint provides only the pre-trained language model parameters, we instantiated the `LlamaForSequenceClassification` module, which attaches an additional linear projection, commonly referred to as the *classification head*, on top of the final hidden-state representation. This head, whose weight matrix is registered as `score.weight` (and optionally `score.bias`), is absent from the checkpoint and was therefore initialized with random Gaussian values.

The fine-tuning setup involves training this weight matrix as well as additive weights for the Q, K, and V layers.

With `Q3R` ($\lambda = 0.0001$, target rank $= 4$, for 100 epochs), we achieved an F1 score of 81.89% on the MRPC dataset of the GLUE benchmark, whereas full fine-tuning (effectively corresponding to setting $\lambda = 0$) resulted in an F1 score of 80.7% after the same number of epochs.

To further compare our proposed method, we conducted experiemnts on Llama3.2-1B on a subset of the GLUE tasks. Table 4 provides a comparison of our method's performance with dense fine-tuning and LoRA.

We achieved the performance as mentioned in Table 4, with a single value of $\lambda$ without any extensive hyperparameter search. While we acknowledge that better performance for GLUE tasks like RTE can often be obtained by starting from a more finely tuned initialization, these results demonstrate that the proposed method is effective even for the fine-tuning of large-scale LLMs.

Table 4: GLUE Benchmark Results for Llama3.2-1B

| Model | MRPC | SST-2 | RTE | CoLA | STS-B |
|-------|------|-------|-----|------|-------|
| Dense | 86.3 | 95.3 | 77.3 | 47.09 | 90.5 |
| LoRA rank=4 | 87.3 | 95.7 | 80.9 | 61.8 | 89.1 |
| Q3R rank=4 | 87.8 | 94.4 | 64.7 | 51.8 | 87.04 |

## D.3 Appendix Tables

Table 5 and Table 6 demonstrate the low rank induction techniques on ViT models. We train ViT-B on CIFAR-100 in Table 6 and ViT-T on CIFAR-10 in Table 5 respectively.

Table 5: Performance at varying percentages of parameters saved on ViT-T when regularizer is applied to only attention blocks.

| Model | 15% | 20% | 30% | 40% | 50% | 60% | 70% | 80% | 90% | 100% |
|-------|-----|-----|-----|-----|-----|-----|-----|-----|-----|------|
| Vanilla ViT-T | 0.4687 | 0.5430 | 0.5892 | 0.6170 | 0.6542 | 0.6694 | 0.6726 | 0.6788 | 0.6892 | 0.7027 |
| Q3R rank=4, $\lambda$=0.01 | 0.6032 | 0.6630 | 0.7043 | 0.7060 | 0.7084 | 0.7080 | 0.7079 | 0.7079 | 0.7077 | 0.7152 |
| Q3R rank=9, $\lambda$=0.01 | 0.4852 | 0.5616 | 0.6668 | 0.6772 | 0.6831 | 0.6835 | 0.6832 | 0.6829 | 0.6831 | 0.7034 |
| Q3R rank=14, $\lambda$=0.01 | 0.4576 | 0.5812 | 0.6866 | 0.6970 | 0.6988 | 0.7002 | 0.6997 | 0.6996 | 0.7003 | 0.7104 |
| Q3R rank=19, $\lambda$=0.01 | 0.6208 | 0.6792 | 0.6913 | 0.6977 | 0.6978 | 0.6970 | 0.6970 | 0.6967 | 0.6966 | 0.7061 |
| Q3R rank=4, $\lambda$=0.001 | 0.5694 | 0.6629 | 0.7017 | 0.7053 | 0.7084 | 0.7091 | 0.7086 | 0.7086 | 0.7079 | 0.7158 |
| Q3R rank=9, $\lambda$=0.001 | 0.4214 | 0.5304 | 0.6467 | 0.6680 | 0.6691 | 0.6692 | 0.6691 | 0.6696 | 0.6695 | 0.6955 |
| Q3R rank=14, $\lambda$=0.001 | 0.6040 | 0.6364 | 0.6642 | 0.6695 | 0.6695 | 0.6694 | 0.6698 | 0.6701 | 0.6704 | 0.6819 |
| LoRITa D=1, $\alpha$=0.1 | 0.1154 | 0.1217 | 0.1234 | 0.1091 | 0.1502 | 0.2310 | 0.3377 | 0.4197 | 0.5027 | 0.7061 |
| LoRITa D=1, $\alpha$=0.001 | 0.1576 | 0.1512 | 0.1602 | 0.1586 | 0.1902 | 0.2456 | 0.2955 | 0.3221 | 0.4056 | 0.7086 |
| LoRITa D=1, $\alpha$=0.01 | 0.1474 | 0.1514 | 0.1539 | 0.1644 | 0.2486 | 0.2968 | 0.4150 | 0.4857 | 0.5326 | 0.6911 |
| LoRITa D=2, $\alpha$=0.01 | 0.2478 | 0.2559 | 0.3115 | 0.4065 | 0.5127 | 0.5641 | 0.5997 | 0.6456 | 0.6720 | 0.7287 |
| LoRITa D=2, $\alpha$=0.1 | 0.2041 | 0.2589 | 0.3639 | 0.5397 | 0.5982 | 0.6408 | 0.6727 | 0.6910 | 0.7025 | 0.7595 |
| LoRITa D=2, $\alpha$=0.001 | 0.2195 | 0.2774 | 0.4163 | 0.5115 | 0.5461 | 0.5675 | 0.6137 | 0.6417 | 0.6614 | 0.7393 |
| LoRITa D=3, $\alpha$=0.01 | 0.3794 | 0.5074 | 0.5907 | 0.6385 | 0.6598 | 0.6761 | 0.6843 | 0.6916 | 0.6959 | 0.7462 |
| LoRITa D=3, $\alpha$=0.001 | 0.3951 | 0.5147 | 0.5952 | 0.6308 | 0.6639 | 0.6676 | 0.6798 | 0.6847 | 0.6833 | 0.7367 |
| LoRA rank=4 | 0.3443 | 0.3443 | 0.3443 | 0.3443 | 0.3443 | 0.3443 | 0.3443 | 0.3443 | 0.3443 | 0.3443 |
| LoRA rank=14 | 0.5859 | 0.5859 | 0.5859 | 0.5859 | 0.5859 | 0.5859 | 0.5859 | 0.5859 | 0.5859 | 0.5859 |

Table 6: CIFAR-100 Performance with ViT-B Attention Block Truncation

| Model | 5% | 10% | 15% | 20% | 30% | 40% | 100% |
|-------|-----|-----|-----|-----|-----|-----|------|
| Vanilla ViT-B Best | 0.2773 | 0.3355 | 0.3659 | 0.3909 | 0.4172 | 0.4327 | **0.4686** |
| Q3R, rank=19, $\lambda$=0.01 | 0.3238 | **0.3979** | **0.4172** | **0.4301** | **0.4411** | **0.4485** | 0.4625 |
| Q3R, rank=19, $\lambda$=0.001 | 0.3174 | 0.3790 | 0.3945 | 0.4050 | 0.4130 | 0.4197 | 0.4408 |
| Q3R, rank=14, $\lambda$=0.001 | 0.3250 | 0.3978 | 0.4149 | 0.4240 | 0.4351 | 0.4429 | 0.4613 |
| Q3R, rank=14, $\lambda$=0.01 | **0.3465** | 0.3950 | 0.4062 | 0.4172 | 0.4305 | 0.4389 | 0.4526 |
| LoRITa D=2, $\alpha$=0.1 | 0.0152 | 0.0444 | 0.0883 | 0.1404 | 0.2428 | 0.3043 | 0.4021 |
| LoRITa D=3, $\alpha$=0.001 | 0.0607 | 0.1375 | 0.2001 | 0.2380 | 0.2975 | 0.3269 | 0.4103 |
| LoRITa D=3, $\alpha$=0.1 | 0.0570 | 0.1369 | 0.2003 | 0.2401 | 0.2946 | 0.3188 | 0.4111 |

## D.4 Computational Aspects

For few experiments like `Q3R`, we used NVIDIA A5000 to train the ViT models. The rest of the experiments were performed on NVIDIA V100 with 32GB memory. The fine-tuning experiemtns were all performed in NVIDIA A5000 GPUs.

## D.5 Computational Overhead of Methodology

In Algorithms 3 and 4, we provide the detailed stepwise FLOP count of our method. Following the experiments testing the influence of the reweighting period on the truncation, we report the average training time of the first 5 epochs below. Note that regularization was only applied on the QKV matrices of a ViT-Tiny Transformer.

Table 7: Average training time (5 epochs) and reserved GPU memory for ViT-Tiny with QKV-only regularization.

| Model | Variant / Setting | Avg. Time (s) | Reserved GPU Mem (GB) |
|---|---|---|---|
| Base | — | 150.91 | 6.26 |
| AdamQ3R | $T=500$ | 205.34 | 6.28 |
| | $T=100$ | 207.39 | 6.31 |
| | $T=50$ | 209.29 | 6.62 |
| | $T=20$ | 217.90 | 8.22 |
| | $T=10$ | 218.88 | 8.62 |
| LoRA | $R=4$ | 124.83 | 11.89 |
| | $R=14$ | 144.51 | 11.87 |
| | $R=28$ | 145.28 | 11.89 |
| Depth1 baseline | — | 157.29 | 8.35 |

Based on the theoretical computational overhead outlined in Algorithm 3 Algorithm 1, along with the results above, we expect Q3R to incur additional computational and memory overhead. The added memory originates from the additional weight matrices stored in Q3R to perform the weight least squares minimization. We attribute the lower Reserved GPU Memory from implementation differences in the optimizer.

# E    Ablation Studies

Table 8 presents the comparison between the regularization term (6) and Algorithm 2 evaluated across varied truncation levels and hyperparameters. Initially AdamQ3R presents competitive performance against Q3R; however, Q3R provides superior truncation performance on the validation set at lower truncation values (20% and below). The impact of the regularization parameter is notably small, with $\lambda = 0.001$ generally providing superior performance.

Table 8: Performance of Vit-T with AdamQ3R and Q3R across different truncation level on CIFAR10.

| Model Name | 5% | 10% | 15% | 20% | 30% | 40% | 100% |
|---|---|---|---|---|---|---|---|
| AdamQ3R, rank = 0.05* | 0.1904 | 0.4600 | 0.6032 | 0.6630 | 0.7043 | 0.7060 | 0.7152 |
| AdamQ3R, rank = 0.10* | 0.2081 | 0.3591 | 0.4852 | 0.5616 | 0.6668 | 0.6772 | 0.7034 |
| AdamQ3R, rank = 0.15* | 0.1266 | 0.3970 | 0.4576 | 0.5812 | 0.6866 | 0.6970 | 0.7104 |
| AdamQ3R, rank = 0.20* | 0.1740 | 0.4717 | 0.6208 | 0.6792 | 0.6913 | 0.6977 | 0.7061 |
| AdamQ3R, rank = 0.05^ | 0.2701 | 0.4608 | 0.5694 | 0.6629 | 0.7017 | 0.7053 | **0.7158** |
| AdamQ3R, rank = 0.10^ | 0.1066 | 0.1741 | 0.4214 | 0.5304 | 0.6467 | 0.6680 | 0.6955 |
| AdamQ3R, rank = 0.15^ | 0.2467 | 0.4993 | 0.6040 | 0.6364 | 0.6642 | 0.6695 | 0.6819 |
| Q3R, rank = 0.04* | 0.2476 | 0.5171 | 0.6544 | 0.6862 | 0.6884 | 0.6874 | 0.6920 |
| Q3R, rank = 0.09* | 0.1959 | 0.5040 | 0.6789 | 0.6789 | 0.6862 | 0.6870 | 0.6827 |
| Q3R, rank = 0.19* | 0.2774 | 0.4317 | 0.6337 | 0.6697 | 0.6818 | 0.6896 | 0.6901 |
| Q3R, rank = 0.04^ | 0.2828 | 0.4699 | 0.6004 | 0.6569 | 0.6801 | 0.6801 | 0.6934 |
| Q3R, rank = 0.09^ | **0.3202** | **0.5527** | **0.6827** | **0.7024** | **0.7024** | **0.7079** | 0.7081 |
| Q3R, rank = 0.19^ | 0.3119 | 0.4440 | 0.6137 | 0.6670 | 0.6839 | 0.6877 | 0.6885 |

**Legend:** * Regularization parameter $\lambda = 0.01$,    ^ Regularization parameter $\lambda = 0.001$

## E.1    Robustness to Hyperparameter Variations

Empirically, we found Q3R to be quite robust to its hyperparameters within reasonable ranges. Below, we are providing few empirical evidences from Table 1on ViT-Tiny. However, similar results can be conducted across other datasets and backbones as in Table 4 (CIFAR-100 on ViT-Base). Generally, when choosing $\lambda$, a viable rule is easy to tell if the choice of lambda is too small by monitoring if the Q3R value increases within the first few epochs. We recommend a value of $\lambda$ that is slightly larger than the lower bound of the divergence threshold, as determining if the $\lambda$ value is too large remains a challenge.

Table 9: Effect of regularization strength $\lambda$ on accuracy.

| Parameter Retention | Accuracy (%) |
|---|---|
| 1% | 46.08–46.30 |
| 5% | 63.40–63.65 |
| 10% | 66.30–66.40 |
| 25% | 70.68–70.74 |
| 50% | 71.04–71.09 |
| 100% (no trunc.) | 71.95–71.52 |

Beyond the 20% retention point, the absolute accuracy gap never exceeds 0.3 percentage points, confirming that AdamQ3R is largely insensitive to $\lambda$ within the 0.001–0.01 range in this practical operating regime.

Table 10: Effect of target rank $r_{\text{target}}$ on accuracy.

| Retention (%) | Accuracy (%) |
|---|---|
| 1% | 0.5683 |
| 5% | 0.6568 |
| 10% | 0.7068 |
| 25% | 0.6671 |
| 50% | 0.6457 |
| 100% | 0.6663 |

Once 30% of the parameters are retained, the choice of rank changes accuracy by $< 0.3\%$, confirming low sensitivity to rank in this regime. We here scale the target rank by layer dimensions so that a single hyper-parameter $r$ works for networks of any size.

## E.2  Merits of Low-Rank Initialization

In our experiments, we implemented supervised initialization where the regulated weight matrix has a rank greater than or equal to the target rank hyperparameter specified in Q3R. Our findings support the hypothesis that low-rank initialization in Q3R imposes a strong constraint on the optimization landscape when the rank is low, thereby limiting the model's capacity to explore more expensive solutions during training. This hypothesis is further supported by the fine-tuning results on RoBERTa where Q3R is applied to the pretrained model and continuously tuned on the GLUE benchmark tasks. We achieve notable results that support the general intuition that models tend to learn better representation through expensive subspaces during the training process. Due to the time constraint of the evaluation period, we could experiment on only the smaller data we observe that LowRank initialization is unable to surpass the performance of Q3R without such constraint on the initialization. However, the accuracy is within 1 range of the full rank initialized model which proves that our proposed method can be implemented in resource constrained setups as well. We provide the example of the empirical evidence performed on CIFAR-10 on ViT-Tiny with $\lambda = 0.05$ and $\alpha = 0.01$.

## E.3  Choice of Reweighting Period

We observe in Table 12 that higher reweighting periods ($T$) (300, 200, 100) result in underperformance in comparison to the lower reweighting periods($T$) (25, 5). While longer reweighting periods provide some computational performance gains based upon formulation Algorithm 1, we observe superior performance for faster intervals which corresponds to the IRLS-majorisation of the logdet.

**Code**  : The code is available at `https://github.com/ThatE10/q3r.git`.

Table 11: Parameter retention effect on accuracy (AdamQ3R + LowRank).

| Parameter Retention | Accuracy (%) |
|---|---|
| 1% | 46.08 |
| 5% | 63.40 |
| 10% | 66.30 |
| 25% | 70.68 |
| 50% | 71.04 |
| 100% (no trunc.) | 72.19 |

Table 12: Model performance under different truncation percentages. Best value per column is bolded. Each model trained with $\lambda = 0.001, r = 0.2$, Trained for 30 epochs

| Model Name | 5% | 10% | 15% | 20% | 30% | 40% | 100% |
|---|---|---|---|---|---|---|---|
| AdamQ3R, $T = 300$ | 0.2999 | 0.5609 | 0.6651 | 0.6801 | 0.6847 | 0.6838 | 0.6827 |
| AdamQ3R, $T = 200$ | 0.2601 | 0.5914 | 0.6519 | 0.6766 | 0.6869 | 0.6885 | 0.6871 |
| AdamQ3R, $T = 100$ | 0.2764 | 0.4871 | 0.6623 | 0.6776 | 0.6869 | 0.6885 | 0.6936 |
| AdamQ3R, $T = 25$ | **0.3729** | 0.5813 | 0.6555 | 0.6725 | 0.6734 | 0.6778 | 0.6790 |
| AdamQ3R, $T = 5$ | 0.1740 | **0.6838** | **0.6828** | **0.6949** | **0.6995** | **0.7000** | **0.7031** |

