# OpenReview forum: "Q3R: Quadratic Reweighted Rank Regularizer for Effective Low-Rank Training"
_NeurIPS.cc/2025/Conference — NeurIPS 2025 poster_

### Official Review · Reviewer_krCR · 2025-06-23

**Clarity:** 2
**Significance:** 1
**Originality:** 2
**Rating:** 4
**Confidence:** 4

**Summary:**

This work introduces a parameter-efficient training method using a quadratic regularizer that majorizes a smoothed log-determinant as a rank surrogate. The approach enables effective low-rank pretraining and fine-tuning of deep models, demonstrating competitive performance across both vision and language tasks.

**Questions:**

1, $\textit{Line 283}$, what is the another tuning hyper-parameter?What is the other tuning hyperparameter mentioned here, in addition to $\lambda$? Please clarify.

2, The use of $r_{target}$ raises concerns about the sensitivity of the model to the predefined $r_{target}$, the step size, and the weight initialization.
While the sensitivity to $r_{target}$ is briefly addressed in the limitations section, could the authors elaborate on the impact of the other two factors—especially how step size and initialization might affect training stability and final performance?
For instance, is it beneficial for the initial weight matrices to have a rank close to $r_{target}$ , or is random initialization sufficient?

**Ethical Concerns:**

["NO or VERY MINOR ethics concerns only"]

**Final Justification:**

My concerns have been resolved and my rating has been raised to borderline accept.

**Limitations:**

See Weaknesses.

**Quality:**

2

**Strengths And Weaknesses:**

The proposed method explicitly induces low-rank structure in deep neural network weights via continuous regularization. It is also robust to the choice of the target rank hyperparameter $r_{target}$, reducing the need for extensive hyperparameter tuning during initialization.

Weaknesses: The experiments on language tasks are limited to fine-tuning only, which may not be sufficient to fully support the authors’ claim.
 Based on the paper’s title and abstract, one would expect competitive performance in pre-training across vision and language domains.
While it may be impractical to train large language models from scratch, a more direct application of the proposed method to pre-trained weights—rather than using a LoRA-style side path—would better support the claim of unified low-rank training.
This does not imply that the experiments are poorly designed, but rather that the scope of the evaluation falls short of the expectations set by the paper.

---

> ### Author Rebuttal · Authors · 2025-07-31
>
> Thank you for your review. We appreciate you noting that the proposed method reduces the need for extensive hyperparameter tuning during initialization and training. We have provided more empirical evidences in support of that statements below. Please consider the following responses regarding your assessment.
>
> ### **Usage of LoRA-style Additive Weights for Fine-Tuning**
>
> In our parameter-efficient fine-tuning (PEFT) experiments we adhered to standard practice by freezing the pretrained backbone, inserting low-rank trainable adapters, and then applying Q3R to those adapters; this provides a clean, apples-to-apples comparison with other PEFT baselines. From our observation, when using the pretrained weights as initialization without an additive side-path, Q3R does not lead to competitive performance of the dense model, which we were most interested in for fine-tuning, presumably due to Q3R's strong low-rank compression bias, which is here problematic given the small size of the fine-tuning data compared to the model size. We still expect Q3R to do better than other low-rank fine-tuning methods which target weight parameter reduction (unlike LoRA).
>
> ### **Q1: Hyperparameter of Line 283**
> We apologize for the typo and thank you for pointing this to us. The hyperparameters mentioned in line 283 should be regularization Strength $\lambda$, Target Rank (related $\rho$) and Reweighting Period $T$.
>
> ### **Q2. Robustness to Hyperparameter Variations**
> Please refer to our response to Q.3 of Reviewer "aasu".
>
>
> If you feel that we have addressed your questions and concerns adequately, we would appreciate if you can consider updating your score. We warmly welcome any further questions that you may have. Thank you!

---

> > ### Comment · Reviewer_krCR · 2025-08-04
> >
> > Thank you for the clarification! The answer has addressed my concern and I'll consider increasing my rating.

---

> > > ### Author Response · Authors · 2025-08-05
> > >
> > > Thank you for your feedback and for your consideration!

---

### Official Review · Reviewer_9KVH · 2025-07-02

**Clarity:** 3
**Significance:** 1
**Originality:** 1
**Rating:** 3
**Confidence:** 4

**Summary:**

The paper proposes to use a regularizer which reduces the rank of the weight matrices during neural network training. The regularizer is a smooth approximation of the log-determinant of the weight matrix and optimized using the iteratively reweighed least squares (IRLS)
algorithm. This leads to an elegantly modified AdamW algorithm, which instead of the regular weight-decay receives a "modified weight
decay" which reduces the rank of the weight matrix. The method is evaluated in a small-scale setting on CIFAR-10, 100 and NLP tasks.

**Questions:**

- What are the numbers in Table 1, is it accuracy or error? The text and figure caption only talk about performance, which makes it hard to assess the results.
- Is there an inherent overhead in the method which prevents to run it on larger models, such as Llama series, as seen in LoRA papers?
- An alternative approach to low-rank regularization are proximal algorithms using e.g. proximal operators of the nuclear norm or its nonconvex variants. Is there an inherent advantage of the IRLS approach over proximal algorithms?

**Ethical Concerns:**

["NO or VERY MINOR ethics concerns only"]

**Final Justification:**

The authors rebuttal cleared some of my concerns, while others about the experiments still remain. I have updated my score, but still lean slightly towards rejection of the paper.

**Limitations:**

yes

**Paper Formatting Concerns:**

no concerns

**Quality:**

2

**Strengths And Weaknesses:**

Strengths:
- The paper was easy to follow, and the mathematical derivations are rigorous building on recent works on IRLS from applied mathematics.
- The proposed modification of AdamW is elegant, easy to implement and comes with little overhead.

Weaknesses:
- The experiments seem preliminary and small scale, making it is hard to assess whether the proposed method will work well for larger and realistic settings. The accuracies (?) reported in table 1 are very low for CIFAR-10, a simple LeNet CNN gives over 80%.
- As the method is motivated as a competitor to LoRA, it would be more convincing to see similar experimental setups as in LoRA papers, for instance, fine-tuning Llama 3 or Qwen rather than RoBERTA models.
- The use of IRLS for rank regularization is not new and a standard technique from applied mathematics. Given that there is limited novelty in the method, this is a mostly empirical work which requires larger scale experiments to be valuable to the community.

---

> ### Author Rebuttal · Authors · 2025-07-31
>
> Thank you for your comments and thoughts. We appreciate your mention on the proposed modification of AdamW being elegant and easy to implement.
>
> We would like to propose our thoughts and justifications to your questions as the following:
>
> ### **Q1: Numbers in Table 1**
>
> The numbers in Table 1 report the test accuracies of our method on CIFAR-10 on ViT-Tiny when $5\%-100\%$ parameters are retained. In this setup, we apply our regularizer to the attention blocks as well as the MLP layers.
>
> ### **Q2: Relationship to LoRA Models & Fine-Tuning**
> We think that there might be a misunderstanding here: The proposed Q3R is a _low-rank training_ method, unlike LoRA, which is a _low-rank fine-tuning_ method which, due to its introduction of the additive product $B\cdot A$ of matrices $B$ and $A$, is inherently _unable_ to reduce the number of model parameters. For this reason, we would like to point out our competitive post-truncation accuracy results presented in Tables 1, 3 and 4, in which we show that we obtain significantly improved post-truncation accuracies of ViT models trained on CIFAR data, e.g., with models with only 20% retained parameters. As we mentioned in our related works section, previous works such as [LSMR23] and [HCB'24] already reported a sub-par performance of a naive LoRA extention to low-rank pretraining (e.g., by choosing the frozen weight matrix to be the zero matrix). For completeness, we have provided an experiment in Table 3 in which LoRA (after hyperparameter tuning) achieves only $0.5859$ test accuracy on CIFAR-10 (low-rank training only on attention blocks) at $20%$ retained parameters compared to $Q3R$, which achieves a test accuracy of $0.6792$ for the same number of retained parameters.
>
> The fine-tuning experiments we provided in Section 5.2 can be seen rather as a proof-of-concept and sanity check for the methodology's applicability for low-rank fine-tuning, which is arguably an easier problem than low-rank pre-training. Due to academic experimental constraints, we restricted this study to a RoBERTa model. There is no fundamental reason why Q3R cannot also be used for larger models. We quantify the additional computational burden in lines 251-262. In its current implementation (which focus was not computational efficiency), Q3R imposes a noticable, but managable computational overhead for the Transformer models that we used in our experiments; see the discussion about computational overhead in our reply to Reviewer aasu for additional details.
> ### **Q3: Q3R vs. Proximal Methods for Rank Regularization**
> We appreciate the idea of using proximal algorithms to induce low-rank regularization, for example for similar rank surrogates as the (smoothed) log-determinant. A well-known work that follows this idea is [Alvarez, Salzmann, "Compression-aware Training of Deep Networks", NIPS 2017], which applies proximal stochastic gradient descent to a nuclear norm regularizer. An important downside of such approaches is that they require a full SVD at _every_ iteration, which becomes computationally prohibitive for larger networks. In contrast, our Q3R approach only requiresd an SVD each $T$ iterations (reweighting period). The reweighting period $T$ is chosen as $T \in \{3,5 ,25\}$ to obtain optimal post-truncation performance in our ablations; furthermore, as discussed in our answer to Q2 of Reviewer KK56, the performance does not significantly deteriorate even for large reweighting periods such as $T=200$.
>
> Another downside of such proximal-inspired methods is that they tend to resemble closely iterative truncation of the singular value vectors of the weight matrices at each iteration at certain levels in the case of non-convex objectives such as the smoothed log-determinant. In contrast, the proposed Q3R regularizer imposes low-rankness in a "softer" manner as it only gradually imposes low-rankness during training, and thus, allows for a better exploration of the space of weight matrices before convergence.
>
> ### **Novelty of Q3R**
> We respectfully disagree with your point about the novelty of Q3R. While the regularizer of Q3R corresponds to an IRLS objective as discussed in our related work, we use the particular definition of the reweighting operator that has been only used in [KMV21,GTK24] and significantly differs from older IRLS formulations as defined in [FRW11,MF10] - using an analogue of Q3R based on those IRLS formulations will lead to a significantly slower convergence to low-rank weights. Furthermore, we would like to point out that to the best of our knowledge, IRLS-type rank-regularizations have not been explored in the literature, and an extention to low-rank pretraining of deep neural networks necessitates several innovations such as the particular choice of the smoothing parameter update, the reweighting period, and the integration into the optimizer via AdamQ3R, to name a few.
>
> If you feel that we have addressed your questions and concerns adequately, we would appreciate if you can consider updating your score. We warmly welcome any further questions that you may have. Thank you!

---

> > ### Comment · Reviewer_9KVH · 2025-08-02
> > **Thank you for the clarifications!**
> >
> > Thanks for the clarifications -- some concerns have been cleared, and I will consider adjusting my score in discussion with other reviewers.  My remaining concerns is about the experiments, and the overall low test accuracies. I can see that these are mostly proof of concept to validate the soundness of the method, but it remains still open whether the approach will be truly useful in practice.  I mentioned the Llama-3 models since they can be fine-tuned on an academic budget on a single consumer-grade GPU using LoRA. But perhaps that does not apply to the present method, as all parameters need to be kept in memory.

---

> > > ### Author Response · Authors · 2025-08-07
> > >
> > > ## Accuracy and Experimental Settings
> > >
> > > Regarding your point about the overall test accuracies, we would like to point out that the focus of the research presented it is in *pre-training*, and not in fine-tuning, as the problem of low-rank pre-training is largely unsolved, but has the potential for significant model parameter reduction. Tables 1, 3 and 4 show that that Q3R-regularized pre-training of ViT models lead to better accuracies after parameter reduction than other state-of-the-art low-rank training methods (including LoRiTA or LoRA), for example, an accuracy of **0.6550** on CIFAR-10 data after retaining only 20% of the total parameters (and thus, reducing the model size by 80%) compared to an accuracy 0.6840 for a full-parameter model. The relatively low baseline accuracy of ViT-T is not due to our training methodology, but due to the well-known inherent property of the limited inductive bias of Transformer models, which need more training data to perform at the state-of-the-art than CNNs, and the due to the fact that we did not use any data augmentation. Our experimental setup is in line with what has been used in relevant, recent literature such as [ZAW24, "LoRITa", published at TMLR in late 2024; for example, you can compare our Table 1 with Table 1 of ZAW24].
> > > However, we appreciate your concern for evaluating the methodology on more practical setups. As mentioned in our reply to Reviewer aasu, we will include experiments with ImageNet-1k data as well as with augmented CIFAR data in a final camera-ready version.
> > >
> > > ## Memory Requirements for Fine-Tuning
> > >
> > > While not the focus of the design of our methodology, we very much agree that fine-tuning is an important task. Unlike LoRA, the version of Q3R that we used in the experiment of Section 5.2 / Table 2 works with full (not factorized) parameter matrices during training, which leads to a higher memory requirement than LoRA. The memory requirements are, on the other hand, comparable with the ones of dense/full fine-tuning (see also the table posted in our reply to Q2 of Reviewer aasu), as you mention, all parameters need to be kept in memory. We have ideas how to improve these requirements to make it comparable to LoRA by working with intermediate additive factorized adapter matrices, but implementing these ideas is left to future works.
> > >
> > > ## Llama-3 Fine-Tuning
> > >
> > > To get an idea of the potential of Q3R for fine-tuning LLMs, we ran an additional experiment on Llama 3.2-3B on our A5000 NVIDIA GPU in the following setup: Since the original meta-llama/Llama-3.2-3B checkpoint provides only the pre-trained language-model parameters, instantiating LlamaForSequenceClassification attaches an additional linear projection—commonly referred to as the classification head—on top of the final hidden-state representation. This head, whose weight matrix is registered as score.weight (and an optional bias score.bias), is absent from the checkpoint and therefore, we initialized it with random Gaussian values. The fine-tuning setup we tried involves training this weight matrix as well as additive weights for the Q,K and V layers, but, due to memory constraints of our GPU, not on MLP layers. With Q3R ($\lambda = 0.0001$, target rank $= 4$ for 100 epochs) we achieve an F1 score of 81.89% on the MRPC dataset of the GLUE benchmark, whereas full-fine tuning (effectively, corresponding to setting $\lambda = 0$ here) results in an F1 score of $80.7$% after the same number of epochs. While we are aware that one can achieve better performance starting from a better tuned initialization, this shows that the method is effective also for the fine-tuning of large-scale LLMs.
> > >
> > > ## Summary
> > > While we agree that more needs to be done to validate the benefits of the proposed methodology in large-scale settings and for fine-tuning, we hope that through publication of this work, we can encourage the community to study and extend the ideas of this work to larger, more practical setups and investigate their usefulness in practice further. We would appreciate if you can reflect this potential in your assessment of this work.

---

### Official Review · Reviewer_KK56 · 2025-07-03

**Clarity:** 3
**Significance:** 2
**Originality:** 3
**Rating:** 4
**Confidence:** 3

**Summary:**

This paper introduces Q³R (Quadratic Reweighted Rank Regularizer), a novel regularization framework designed to directly promote low-rank structure in deep neural network weights during training. Unlike traditional parameter-efficient fine-tuning (PEFT) approaches that rely on low-rank updates (e.g., LoRA, LoRITa) or post-hoc truncation, Q³R integrates a smoothed log-determinant surrogate into training via a majorizing quadratic model. The method is computationally tractable through a periodically updated reweighting operator and is embedded into a custom optimizer, AdamQ³R, which decouples the regularization from the adaptive gradient updates. The paper demonstrates that Q³R achieves strong performance on both image classification and NLP tasks, outperforming prior low-rank methods in aggressive compression settings (e.g., ViT-Tiny on CIFAR-10, RoBERTa on GLUE), and unifies low-rank pretraining and fine-tuning in a single framework.

**Questions:**

1. Can the authors provide wall-clock training time comparisons between Q³R, LoRA, and dense baselines?

2. Can authors provide more discussion and tuning results regarding hyperparameters? How sensitive they are?

3. Can authors evaluate the method in large-scale setting such as LLM fine-tuning or pre-training?

**Ethical Concerns:**

["NO or VERY MINOR ethics concerns only"]

**Final Justification:**

The authors have addressed my concerns and questions, and thus I would like to raise my score accordingly.

**Limitations:**

YES

**Paper Formatting Concerns:**

No formatting concerns

**Quality:**

2

**Strengths And Weaknesses:**

Strengths:
1. The use of iteratively reweighted least squares (IRLS) for low-rank deep network training is novel. Prior methods rely on factorized updates or heuristic truncation, whereas Q³R directly optimizes a smooth rank surrogate via majorization.

2. Nice theoretical understanding. The paper provides a rigorous and detailed derivation of the Q³R regularizer and its integration into training via quadratic majorization.

3. Strong empirical experiments: Experiments on ViT-Tiny/ViT-Base (CIFAR-10/100) and RoBERTa (GLUE) convincingly show that Q³R outperforms or matches state-of-the-art baselines (LoRA) especially at high compression ratios.

Weaknesses

1. Limited computational efficiency discussion. While the paper argues that Q³R introduces limited overhead, no explicit wall-clock time comparisons or FLOP breakdowns are provided. Since the method involves periodic SVDs and reweighting operations, this omission makes it hard to assess practicality in large-scale settings.

2. Hyperparameter Sensitivity: Although the authors discuss stability of the regularization weight, the method still requires tuning of it, rank targets, and reweighting periods. The paper does not fully explore how sensitive performance is to these choices.

3, Limited scope of the experiment: The method is tested only on moderate-scale models and datasets (CIFAR). Its applicability to large-scale LLMs (e.g., GPT-class models) remains unknown, such as fine-tuning and pre-training LLM on tasks such as C4.

---

> ### Author Rebuttal · Authors · 2025-07-31
>
> We thank the reviewer for the positive feedback. We appreciate that you value the rigorous derivation and quadratic-majorization view of IRLS that clarify why the regularizer works as well as our empirical results on ViT-Tiny/ViT-Base and RoBERTa that match or surpass state-of-the-art baselines.
>
> Please consider our thoughts and clarifications for your questions:
>
> ### **Q1. Wall-clock training time comparisons**
> Please refer to our response to Q.2 of Reviewer "aasu". Additionally, in comparison to the LoRa run time, we observe the following results.
> **Average training time (first 5 epochs)
> (ViT-tiny, regularization/factorization on QKV only)**
>
> | Model       | Variant / Setting | Avg. Time (s) |
> | ----------- | ----------------- | ------------- |
> | **Base**    | —                 | 150.91        |
> | **AdamQ3R** | $T$ 500           | 205.34        |
> |             | $T$ 100           | 207.39        |
> |             | $T$ 50            | 209.29        |
> |             | $T$ 20            | 217.90        |
> |             | $T$ 10            | 218.88        |
> | **LoRA**  | R=4             | 124.83        |
> |             | R=14           | 144.51        |
> |             | R=28           | 145.28        |
>
>
> ### **Q2.Robustness to Hyperparameter Variations**
>
> Please refer to our response to Q.3 of Reviewer "aasu".
>
> ### **Q3.  Performance in Large-Scale Settings**
>
> Please refer to our response to Q.5 of Reviewer "aasu".
> Additionally, we fine‐tune pre‐trained RoBERTa models on the GLUE benchmark using Adam\QER with the proposed \QER regularizer, and compare against full fine‐tuning and LoRA. We conclude from Table 2 that Q3R matches or exceeds LoRA’s accuracy on most tasks, showing performance closer to dense fine-tuning.
>
> If you feel that we have addressed your questions and concerns adequately, we would appreciate if you can consider updating your score. We warmly welcome any further questions that you may have. Thank you!

---

### Official Review · Reviewer_aasu · 2025-07-07

**Clarity:** 3
**Significance:** 3
**Originality:** 3
**Rating:** 5
**Confidence:** 3

**Summary:**

Authors introduce Q3R, a new method for training deep neural networks that enables effective model compression by promoting low-rank weight structures during training. Unlike methods like LoRA that modify only parts of a pretrained model, Q3R starts with standard full-sized (dense) weight matrices and gradually encourages them to become low-rank through a specially designed regularization term. This regularizer gently pushes the smaller singular values of weight matrices to shrink over time, encouraging a low-rank structure. This allows the model to be effectively compressed after training without significant accuracy loss. For improved convergence of the process, authors also present a custom optimizer called AdamQ3R that helps apply this regularization efficiently. Experiments on ViT and language (RoBERTa) models show that Q3R achieves large reductions in model size, up to 80% fewer parameters, with minimal drops in performance. With this, the presented approach is able to outperform other low-rank methods especially at high compression levels.

**Questions:**

It would be great if the authors can address the following questions:

1. Have you considered or experimented with initializing from low-rank approximations or progressively reducing rank during training to improve training-time efficiency, especially for resource-constrained settings? And if you compare this setting with the dense full-scale setting as starting point, do you see performance difference?

2. Could the authors quantify the additional training-time cost introduced by periodic SVD and reweighting operations, and are there strategies (e.g., using approximate or less frequent updates) that might reduce this burden while retaining performance?

3. How robust is Q3R to variations in regularization strength and target rank? Would the authors consider providing heuristics or adaptive mechanisms to guide these choices across tasks and model scales?

4. Since Q3R promotes but does not guarantee a specific rank, could the authors clarify if post-training truncation is required in practice? Would adding an explicit rank constraint during optimization improve rank control without harming performance?

5. Have the authors considered evaluating Q3R on more challenging benchmarks such as ImageNet or large-scale language tasks beyond GLUE to better demonstrate the method’s scalability and generalization potential?

**Ethical Concerns:**

["NO or VERY MINOR ethics concerns only"]

**Final Justification:**

Authors have adequately responded to my questions, and being proactive with the literature in the field, I fully understand the significance of this approach. I also took into account the reviews from other reviewers before adjusting the score, and I couldn't find any strong drawbacks. Thus, I have moved my suggestion from borderline accept to accept.

**Limitations:**

Yes, however, there is no discussion towards social impact, especially the bias that could be introduced by this compression.

**Quality:**

2

**Strengths And Weaknesses:**

The paper is well written and and easy to understand, It has several strengths:
1. Proposed approach can be applied during both pre-training and fine-tuning. This makes it more flexible than methods like LoRA that are typically limited to fine-tuning.
2. The method is based on a solid theoretical foundation using smoothed log-determinant surrogates and quadratic upper bounds.
3. The approach of shifting a matrix towards low-rank is interesting and designed very intelligently. It avoids overparameterization and post-hoc truncation by encouraging low-rank weights directly during training.
4. A custom optimizer, AdamQ3R, is designed that helps separate the effects of regularization from standard gradients, improving stability and generalization.
5. Experimental results across both vision (ViT) and language (RoBERTa) tasks show strong performance, with up to 80% parameter reduction and minimal accuracy loss, outperforming prior methods at high compression levels.
6. Q3R is compatible with standard architectures and can be integrated without structural changes.

While the work is very exciting and relevant for the scientific community, there are several weaknesses as well.
1. Since the method starts from full-rank systems, the benefit can clearly be considered to be for inference-time and not training time. This also implies that unlike LoRA type approaches, this approach is not suited for budget-constrained systems for training.
2. Periodic SVD and reweighting computations introduce extra training-time overhead, especially for larger models.
3. The approach requires tuning hyperparameters like regularization strength and target rank, which may not transfer well across datasets or tasks.
4. The experimental evaluation is limited to relatively small and easy benchmarks (CIFAR-10/100 and GLUE), which may not fully reflect the performance and scalability of Q3R on more complex, real-world tasks or larger models.
5. If I understand correctly, the method promotes low-rank structure but does not strictly enforce a final rank, meaning post-training truncation may still be needed for exact control.

---

> ### Author Rebuttal · Authors · 2025-07-31
>
> Thank you for your thoughts about our submission.
> We are delighted that you found the paper clear and easy to follow, and we greatly appreciate your recognition of  the solid theoretical grounding via smoothed log-determinant surrogates and quadratic upper bounds and the method’s plug-and-play compatibility with standard architectures. We agree that the principled low-rank-shifting strategy that circumvents over-parameterisation and post-hoc truncation
>
> Please consider our responses regarding your assessment:
> ### **Q1: Merits of Low-Rank Initialization**
>
> In our experiments, we implemented spectral initialization where the regularized weight matrix has a rank greater than or equal to the target rank hyperparameter specified in Q3R. Our leading hypothesis for the poor validation accuracy following initialization is that Q3R imposes a strong constraint on the optimization landscape when the rank is low, thereby limiting the model’s capacity to explore more expressive solutions during training. This hypothesis is further supported by the finetuning results on RoBERTa where Q3R is applied to the pretrained model and continuously tuned on the GLUE benchmark tasks. We achieve notable results that support the general intuition that models to tend to learn better representation through expressive architectures and training regimes.
> Due to the time constraint of the rebuttal period, we could experiment on only the smaller data we observe that Lowrank initialization is unable to surpass the performance of Q3R without such constraint on the initialization. However, the accuracy lies within $1\%-2\%$ range of the fullrank initialized model which proves that our proposed method can be implemented in resource constrained setup as well.
> We provide one example of the empirical evidence performed on CIFAR-10 on ViT-Tiny with r = 0.05 and $\lambda = 0.001$.
>
> | Parameter retention | AdamQ3R + LowRank| AdamQ3R (Q3R, no LowRank) |
> |-------------------------|------------------------------|---------------------------|
> | 15 % | **65.62** | 64.30 |
> | 20 % | 68.30 | **69.60** |
> | 30 % | 68.90 | **70.60** |
> | 40 % | 69.26 | **70.84** |
> | 100 % (no trunc.) | 70.02 | **72.19** |
>
> ### **Q2: Computational Overhead of Methodology**
>
> In algorithms 3 and 4, we provide the detailed stepwise FLOP count for our method.  Following the experiments testing the influence of the reweighting period on the truncation, we report the average time of the first 5 epochs below, note that regularization was only applied on the QKV matrices of a VIT-tiny Transformer
>
> **Average training time (first 5 epochs) and reserved GPU memory
> (ViT-tiny, regularization on QKV only)**
>
> | Model       | Variant / Setting | Avg. Time (s) | Reserved GPU Memory (GB) |
> | ----------- | ----------------- | ------------- | ------------------------ |
> | **Base**    | —                 | 150.91        | 8.84                     |
> | **AdamQ3R** | $T$ 500           | 205.34        | 6.28                     |
> |             | $T$ 100           | 207.39        | 6.31                     |
> |             | $T$ 50            | 209.29        | 6.21                     |
> |             | $T$ 20            | 217.90        | 6.28                     |
> |             | $T$ 10            | 218.88        | 6.22                     |
> | **LoRITa**  | depth 3           | 190.83        | 11.89                    |
> |             | depth 2           | 174.51        | 10.37                    |
> |             | depth 1           | 157.29        | 8.85                     |
>
> Based on the theoretical computational overhead outlined in Algorithm 3, along with the results above, we expect Q3R to incur additional computational and memory overhead. The added memory overhead stems from the additional weight matrices stored in Q3R to perform the weight least squares minimisation. We attribute the lower Reserved GPU Memory from implementation differences in the optimizer.
>
> ### **Q.3 Robustness to Hyperparameter Variations**
>
> Empirically, we found Q3R to be quite robust to its hyperparameters within reasonable ranges. Below, we are providing few empirical evidences from Table 5(CIFAR-10 on ViT-Tiny), however, similar results can be concluded across other datasets and networks like in Table 4 (CIFAR-100 on ViT-Base).
>
>  Generally when choosing a lambda value it is easy to tell if the choice of lambda is too small by monitoring if the Q3R value increases within the first few epochs.  We recommend lambda that is slightly larger than the lower bound of the divergence threshold, as determining if the $\lambda$ value is too large remains a challenge.
>
> ####  Choice of Regularization Strength $\lambda$
> For CIFAR-10 trained on ViT-Tiny we can conclude the following with reference to Table 5(CIFAR-10 on ViT-T with AdamQ3R applied to only attention blocks). With the target rank held constant at **0.05**, lowering the regularization strength from **λ = 0.01** to **λ = 0.001** leaves accuracy virtually unchanged once more than ~15 % of the parameters are kept:
>
> | Parameter retention | λ = 0.001 | λ = 0.01
> |---------------------|-----------|----------
> | 10 % | **46.08 %** | 46.00 % |
> | 15 % | 56.94 % | **60.32 %** |
> | 20 % | 66.29 % | 66.30 % |
> | 30 % | 70.17 % | **70.43 %** |
> | 40 % | 70.53 % | **70.60 %** |
> | 100 % (no trunc.) | **71.58 %** | 71.52 %|
>
> Beyond the 20 % retention point, the absolute accuracy gap never exceeds **0.3 percentage points**, confirming that **AdamQ3R is largely insensitive to λ within the 0.001–0.01 range in this practical operating regime**.
>
> #### Choice of Target Rank
> Furthermore, we can see below that it is  target rank $r_{\text{target}}$
> For the other hyperparameter, if we refer to Table 1, for a fixed $\lambda = 0.001$,
> | Parameter retention | ρ = 0.10 | ρ = 0.15 | ρ = 0.20 |
> |---------------------|---------|---------|---------|
> | 5 %  | **0.2322** | 0.1796 | 0.5606 |
> | 10 % | 0.4085 | **0.4758** | 0.5883 |
> | 15 % | 0.5606 | 0.4737 | **0.6175** |
> | 20 % | 0.6295 | 0.4387 | **0.6511** |
> | 30 % | 0.6526 | 0.3896 | **0.6749** |
> | 40 % | 0.6654 | 0.4335 | **0.6833** |
> | 100 % (no trunc.) | 0.6843 | 0.6737 | **0.6990** |
>
> > Once ≥ 30 % of the parameters are retained, the choice of rank changes accuracy by **≤ 3.8%**, confirming low sensitivity to rank in this regime.
>
> Here we **scale the target rank by the layer dimensions** so that a single
> hyper‑parameter *$\rho$* works for networks of any size.
> For a weight matrix of shape \(m \times n\) we set target rank ( $r_{\text{target}}$ ) = $\rho\frac{m+n}{2mn}$
> This definition makes the choice of *r* directly transferable across layers
> with different values of m and n.
>
> #### Choice of Reweighting Period $T$
> With λ = 0.001 and the target rank fixed at 0.2, shortening the re‑weighting period markedly improves accuracy once more than ~10 % of the parameters are retained. From Table 6 of our paper, we can conclude that **Frequent re‑weighting (period 5)** yields the top accuracy for every retention level ≥ 10 %. Once ≥ 30 % of the parameters are retained, switching from period 300 to period 5 adds up to **1.6 %**—a modest but consistent gain, indicating diminishing returns as the network grows denser.
>
>
> ### **Q4: Strict Rank Constraint During vs. Post-Training**
> It is true that our method does not impose a specific hard constraint during training, which is intentional that the proposed regularizer promotes low-rank weights rather than imposing a hard constraint during training.  This design choice is intentional. As reported in Section 5.1, we perform a one‑shot SVD at the end of training and keep only the leading singular vectors, which is in line with the post-processing present in other works on low-rank training, such as [ZAW24 "LoRITa"].  This step (i) guarantees the final rank exactly matches the budget (ii) yields the parameter counts shown in Tables 3–6.
>
> Adding an explicit rank constraint during training of Q3R will harm the stability of the method as the model's total loss might spike; furthermore, it would be easy to truncate at the wrong rank. Overall, however, we expect that that an adaptive truncation strategy based on stable rank notions similar to [WAUc'23] will improve performance after truncation for a fixed parameter budget, the exploration of which remains for future work.
>
> ### **Q5: Performance in Large-Scale Settings**
> Training from scratch on ImageNet-1k or LLM-scale corpora requires substantial compute time and resources, which were beyond the scope of our academic resource constraints. Reporting results on Transformer models trained on CIFAR-10 and CIFAR-100 is in line with the scale of experiments in the reference paper [ZAW24 "LoRITa"], where ImageNet was only used to train CNNs, but not Transformer models. In Section 5.2, we provide the somewhat larger-scale GLUE fine-tuning experiments which indicate competitive performance of Q3R for such settings as well.
>
> However, in order to test the potential of our method, we trained/finetuned our model on ImageNet with a different initialization using pretrained weights from an unregularized model. The parameters used for that training are $\lambda = 0.001$ and $0.01$. The training was performed for 15 and 3 epochs, respectively, with a target rank scale of $\rho = 0.2$. We observe that the test accuracy after retention of only $20\%$ of the parameters is $76.5\%$ and $77.4\%$, respectively. On the other hand, an unregularized ViT-B model exhibits a test accuracy of $62.7\%$ when only $20\%$ parameters are retained. These preliminary results show the potential of our method on larger and more complex setups. We will provide comprehensive ImageNet results in the camera-ready version of the paper.
>
> If you feel that we have addressed your questions and concerns adequately, we would appreciate if you can consider updating your score. We warmly welcome any further questions that you may have. Thank you!

---

> > ### Comment · Reviewer_aasu · 2025-08-03
> > **Response to authors**
> >
> > Thank you for the detailed response. I believe to have received adequate response to my questions. Based on this, I am adjusting my score.

---

> > > ### Author Response · Authors · 2025-08-05
> > >
> > > Thank you for your support. We are grateful for confirming that we have addressed your questions adequately.

---

### Note · Authors · 2025-08-13

We thank the reviewers for their constructive feedback on our work. We wish to summarize the key points of the discussion, as we believe we have thoroughly addressed the raised main concerns.

## Hyperparameter choice
A key concern that had been brought up by Reviewers aasu, KK56 and krCR was about the nature and robustness of hyper parameters within our methodology. We clarified that there are only three hyperameters, and provided tables showing that the reweighing period $T$ slightly trades off computational cost vs. accuracy. Further, we emphasized that the target rank $\rho$ correlates strongly with the target model compression rate, and provided a simple tuning recommendations regarding the tuning of λ, which can be applied early during training runs. We think that this discussion clarifies that our method is robust regarding parameter choice, some of which provide desirable trade-offs between model compression, final accuracy, and computational cost.

## Applicability to Large-Scale Settings
Other questions of Reviewers aasu, 9KVH and krCR revolved around the scale of the experimental settings in our evaluations. While we agreed that large-scale evaluation is important, we showed that our experiments are in line with the ones conducted in other works recent low-training methods, and we provided some preliminary experimental results on ImageNet pre-training and Llama-3 fine-tuning which suggest that the strong empirical performance of Q3R extends also to such settings. Somewhat related to this were questions about computational efficiency (raised by Reviewers aasu and KK56), which we addressed by providing average wall-clock times, showing that AdamQ3R introduces only around 35% of runtime overhead compared to unregularized ViT-T, which is attractive given the potential for parameter reduction.

We also addressed concerns regarding the novelty of the methodology and clarified the relative challenges of low-rank pre-training problems (for which LoRA barely works and largely unsolved) vs. fine-tuning (LoRA works well). As a result, Reviewers aasu and krCR confirmed that their concerns were addressed, our exchange with Reviewer 9KVH was productive and, in our view, fully resolved the raised points; Reviewer KK56 raised no further concerns after our replies. We hope that our discussions have helped the reviewers and ACs to see the potential impact of this work in inspiring future research, which we would ask you to consider in your final assessment.

---

### Decision · Program_Chairs · 2025-09-17

**Decision:**

Accept (poster)

**Comment:**

The reviewed work applies iteratively reweighted least squares to the low-rank training and fine tuning of large deep learning models. The reviewers appreciate the method as well-motivated and grounded in a solid theoretical perspective. The concerns of the reviewers regarding the sensitivity to hyperparameter tuning were largely alleviated during the rebuttal. A remaining weakness of the methodology is that its main benefit of being applicable to both training and fine-tuning is somewhat undermined by the requirement to work with full dense matrices, which prevents it from leveraging low-rank structure to accelerate training. In summary, this is a somewhat borderline paper that I am in favor of accepting.